# Robustness of Community Detection to Random Geometric Perturbations

**Sandrine Péché**
LPSM, University Paris Diderot
peche@lpsm.fr

**Vianney Perchet**
Crest, ENSAE & Criteo AI Lab
vianney.perchet@normalesup.org

## Abstract

We consider the stochastic block model where connection between vertices is perturbed by some latent (and unobserved) random geometric graph. The objective is to prove that spectral methods are robust to this type of noise, even if they are agnostic to the presence (or not) of the random graph. We provide explicit regimes where the second eigenvector of the adjacency matrix is highly correlated to the true community vector (and therefore when weak/exact recovery is possible). This is possible thanks to a detailed analysis of the spectrum of the latent random graph, of its own interest.

## Introduction

In a $d$-dimensional random geometric graph, $N$ vertices are assigned random coordinates in $\mathbb{R}^d$, and only points close enough to each other are connected by an edge. Random geometric graphs are used to model complex networks such as social networks, the world wide web and so on. We refer to [19] - and references therein - for a comprehensive introduction to random geometric graphs. On the other hand, in social networks, users are more likely to connect if they belong to some specific community (groups of friends, political party, etc.). This has motivated the introduction of the stochastic block models (see the recent survey [1] and the more recent breakthrough [5] for more details), where in the simplest case, each of the $N$ vertices belongs to one (and only one) of the two communities that are present in the network.

The two types of connections – geometric graph vs. block model – are conceptually quite different and co-exist independently. Two users might be connected because they are "endogenously similar" (their latent coordinates are close enough to each others) or because they are "exogenously similar" (they belong to the same community). For instance, to oversimplify a social network, we can consider that two different types of connections can occur between users: either they are childhood friends (with similar latent variables) or they have the same political views (right/left wing).

We therefore model these simultaneous types of interaction in social networks as a simple stochastic block model (with 2 balanced communities) perturbed by a latent geometric graph. More precisely, we are going to assume that the probability of endogenous connections between vertices $i$ and $j$, with respective latent variables $X_i, X_j \in \mathbb{R}^d$, is given by the Gaussian[1] kernel $\exp(-\gamma \|X_i - X_j\|^2)$ where $\gamma$ is the (inverse) width. On the other hand, exogenous connections are defined by the block model where half of the $N$ vertices belong to some community, half of them to the other one. The probability of connection between two members of the same community is equal to $p_1$ and between two members from different communities is equal to $p_2$. We also consider an extra parameter

$\kappa \in [0,1]$ to represent the respective strengths of exogenous vs. endogenous connections (and we assume that $\kappa + \max\{p_1, p_2\} \leq 1$ for technical reason).

Overall, the probability of connection between $i$ and $j$, of latent variable $X_i$ and $X_j$ is

$$\mathbb{P}\{i \sim j \mid X_i, X_j\} = \kappa e^{-\gamma\|X_i - X_j\|^2} + \begin{cases} p_1 & \text{if } i, j \text{ are in the same community} \\ p_2 & \text{otherwise} \end{cases}$$

In stochastic block models, the key idea is to recover the two communities from the observed set of edges (and only from those observations, i.e., the latent variables $X_i$ are not observed). This recovery can have different variants that we enumerate now (from the strongest to the weakest). Let us denote by $\sigma \in \{\frac{\pm 1}{\sqrt{N}}\}^N$ the normalized community vector illustrating to which community each vertex belong ($\sigma_i = -\frac{1}{\sqrt{N}}$ if $i$ belongs the first community and $\sigma_i = \frac{1}{\sqrt{N}}$ otherwise).

Given the graph-adjacency matrix $A \in \{0,1\}^{N^2}$, the objective is to output a normalized vector $x \in \mathbb{R}^N$ (i.e., with $\|x\| = 1$) such that, for some $\varepsilon > 0$,

**Exact recovery:** with probability tending to 1, $|\sigma^\top x| = 1$, thus $x \in \{\frac{\pm 1}{\sqrt{N}}\}^N$

**Weak recovery:** with probability tending to 1, $|\sigma^\top x| \geq \varepsilon$ and $x \in \{\frac{\pm 1}{\sqrt{N}}\}^N$

**Soft recovery:** with probability tending to 1, $|\sigma^\top x| \geq \varepsilon$

We recall here that if $x$ is chosen at random, independently from $\sigma$, then $|\sigma^\top x|$ would be of the order of $\frac{1}{\sqrt{N}}$, thus tends to 0. On the other hand, weak recovery implies that the vector $x$ has (up to a change of sign) at least $\frac{N}{2}(1 + \varepsilon)$ coordinates equal to those of $\sigma$. Moreover, we speak of soft recovery (as opposed to hard recovery) in the third case by analogy to soft vs. hard classifiers. Indeed, given any normalized vector $x \in \mathbb{R}^d$, let us construct the vector $\text{sign}(x) = \left(\frac{2\mathbb{1}\{X_i \geq 0\} - 1}{\sqrt{N}}\right) \in \{\frac{\pm 1}{\sqrt{N}}\}^N$. Then $\text{sign}(x)$ is a candidate for weak/exact recovery. Standard comparisons between Hamming and Euclidian distance (see, e.g., [16]) relates soft to weak recovery as

$$|\sigma^\top \text{sign}(x)| \geq 4|\sigma^\top x| - 3;$$

In particular, weak-recovery is ensured as soon as soft recovery is attained above the threshold of $\varepsilon = 3/4$ (and obviously exact recovery after the threshold $1 - 1/4N$).

For simplicity, we are going to assume[2] that $X_i$ are i.i.d., drawn from the 2-dimensional Gaussian distribution $\mathcal{N}(0, I_2)$. In particular, this implies that the law $A_{i,j}$ (equal to 1 if there is an edge between $i$ and $j$ and 0 otherwise) is a Bernoulli random variable (integrated over $X_i$ and $X_j$) $\text{Ber}\left(\frac{p_1 + p_2}{2} + \frac{\kappa}{1 + 4\gamma}\right)$; Notice that $A_{i,j}$ and $A_{i',j'}$ are identically distributed but not independent if $i = i'$ or $j = j'$. Recovering communities can be done efficiently (in some regime) using spectral methods and we will generalize them to this perturbed (or mis-specified) model. For this purpose, we will need a precise and detailed spectral analysis of the random geometric graphs considered (this has been initiated in [20], [10] and [4] for instance).

There has been several extensions of the standard stochastic block models to incorporate latent variables or covariables in perturbed stochastic block models. We can mention cases where covariables are observed (and thus the algorithm can take their values into account to optimize the community recovery) [25, 23, 9, 14], when the degree of nodes are corrected [12] or the case of labeled edges [13, 24, 15, 16, 26]. However, these papers do not focus on the very simple question of the robustness of recovery algorithm to (slight) mis-specifications in the model, i.e., to some small perturbations of the original model and this is precisely our original motivations. Regarding this question, [21] consider the robustness of spectral methods for a SBM perturbed by adversarial perturbation in the sparse degree setup. Can we prove that a specific efficient algorithm (here, based on spectral methods) still exactly/weakly/softly recover communities even if it is agnostic to the presence, or not, of endogenous noise? Of course, if that noise is too big, then recovery is impossible (consider for instance the case $\gamma = 0$ and $\kappa \gg 0$). However, and this is our main contribution, we are able

to pinpoint specific range of perturbations (i.e., values of $\kappa$ and $\gamma$) such that spectral methods – in short, output the normalized second highest eigenvector – still manage to perform some recovery of the communities. Our model is motivated to simplify the exposition but can be generalized to more complicated models (more than two communities of different sizes).

To be more precise, we will prove that:
- if $1/\gamma$ is in the same order than $p_1$ and $p_2$ (assuming that $p_1 \sim p_2$ is a standard assumption in stochastic block model), then soft recovery is possible under a mild assumption ($\frac{p_1 - p_2}{2} \geq 4\frac{\kappa}{\gamma}(1+\varepsilon)$);
- if $\gamma(p_1 - p_2)$ goes to infinity, then exact recovery happens.
However, we mention here that we do not consider the "sparse" case (when $p_i \sim \frac{a}{n}$), in which regimes where partial recovery is possible or not (and efficiently) are now clearly understood [7, 17, 8, 18], as the geometric graphs perturbes too much the delicate arguments.

Our main results are summarised in Theorem 8 (when the different parameters are given) and Theorem 10 (without knowing them, the most interesting case). It is a first step for the study of the robustness of spectral methods in the presence of endogenous noise regarding the question of community detection. As mentioned before, those results highly rely on a careful and detailed analysis of the spectrum of the random graph adjacency matrix. This is the purpose of the following Section 1, which has its own interest in random graphs. Then we investigate the robustness of spectral methods in a perturbed stochastic block model, which is the main focus of the paper, in Section 2. Finally, more detailed analysis, other statements and some proofs are given in the Appendix.

# 1  Spectral analysis for the adjacency matrix of the random grah

Let us denote by $P$ the *conditional expectation matrix* (w.r.t the Gaussian kernel), where $P_{ij} = P_{ji} = e^{-\gamma||X_i - X_j||^2}$, for $i < j \in [1, .., N]$, and $P_{ii} = 0$ for all $i = 1, .., N$. We will denote by $\mu_1 \geq \mu_2 \geq \cdots \geq \mu_N$ its ordered eigenvalues (in Section 2, $\mu_k$ are the eigenvalues of $\kappa P$).

## 1.1  The case where $\gamma$ is bounded

We study apart the case where $\limsup_{N\to\infty} \gamma < \infty$. The simplest case corresponds to the case where $\gamma \log(N) \to 0$ as $N \to \infty$ as with probability one, each $P_{i,j}$ converges to one. And as a consequence, the spectrum of $P$ has a nonzero eigenvalue which converges to $N$ (with probability arbitrarily close to 1). In the case where $\gamma$ is not negligible w.r.t. $\frac{1}{\log(N)}$, arguments to understand the spectrum of $P$ – or at least its spectral radius – are a bit more involved.

**Proposition 1.** *Assume that $\gamma(N)$ is a sequence such that $\lim_{N\to\infty} \gamma(N) = \gamma_0 \geq 0$. Then there exists a constant $C_1(\gamma_0)$ such that the largest eigenvalue of $P$ satisfies*

$$\frac{\mu_1(P)}{NC_1(\gamma_0)} \to 1 \text{ as } N \to \infty.$$

## 1.2  The spectral radius of $P$ when $\gamma \to \infty$, $\gamma \ll N/\ln N$

We now investigate the special case where $\gamma \to \infty$, but when $\gamma \ll N/\ln N$ (as in this regime the spectral radius $\rho(P)$ of $P$ does not vanish). We will show that $\rho(P)$ is in the order of $\frac{N}{2\gamma}$.

We formally state this case under the following Assumption ($H_1$) (implying that $\gamma \ln \gamma \ll N$).

$$\gamma \to \infty \text{ and } \frac{1}{\gamma}\frac{N}{\ln N} \to \infty. \tag{$H_1$}$$

**Proposition 2.** *If Assumption ($H_1$) holds then, with probability tending to one,*

$$\frac{N}{2\gamma} \leq \rho(P) \leq \frac{N}{2\gamma}\left(1 + o(1)\right).$$

*Proof.* By the Perron Frobenius theorem, one has that

$$\min_{i=1,...,N}\sum_{l=1}^{N} P_{il} \leq \rho(P) \leq \max_{i=1,...,N}\sum_{l=1}^{N} P_{il}.$$

To obtain an estimate of the spectral radius of $P$, we show that, with probability tending to 1, $\max_i \sum_{l=1}^{N} P_{il}$ cannot exceed $\frac{N}{2\gamma}$ and for "a large enough number" of indices $i$, their *connectivity* satisfies

$$\sum_{l=1}^{N} P_{il} = \frac{N}{2\gamma}(1 + o(1)).$$

The proof is going to be decomposed into three parts (each corresponding to a different lemma, whose proofs are delayed to Appendix B.).

1. We first consider only vertices close to 0, i.e., such that $|X_i|^2 \leq 2\frac{\log(\gamma)}{\gamma}$. For those vertices, $\sum_j P_{i,j}$ is of the order of $N/2\gamma$ with probability close to 1. See Lemma 3

2. For the other vertices, farther away from 0, it is easier to only provide an upper bound on $\sum_j P_{i,j}$ with a similar proof. See Lemma 4

3. Then we show that the spectral radius has to be of the order $N/2\gamma$ by considering the subset $J$ of vertices "close to 0" (actually introduced in the first step) and by proving that their inner connectivity – restricted to $J$ –, must be of the order $N/2\gamma$. See Lemma 5.

Combining the following three Lemmas 3, 4 and 5 will immediately give the result. $\qquad\square$

**Lemma 3.** *Assume that Assumption ($H_1$) holds, then, as $N$ grows to infinity,*

$$\mathbb{P}\left\{\exists i \leq N \text{ s.t. } |X_i|^2 \leq 2\frac{\ln\gamma}{\gamma}, \left|\sum_{j=1}^{N} P_{ij} - \frac{N}{2\gamma}\right| \leq o\left(\frac{N}{2\gamma}\right)\right\} \to 1.$$

Lemma 3 states that the connectivities of vertices close to the origin converge to their expectation (conditionally to $X_i$). Its proof decomposes the set of vertices into those that are close to $i$ (the main contribution in the connectivity, with some concentration argument), far from $i$ but close to the origin (negligible numbers) and those far from $i$ and the origin (negligible contribution to the connectivity).

The second step of the proof of Proposition 2 considers indices $i$ such that $|X_i|^2 \geq 2\frac{\ln\gamma}{\gamma}$.

**Lemma 4.** *For indices $i$ such that $|X_i|^2 \geq 2\frac{\ln\gamma}{\gamma}$ one has with probability tending to 1 that*

$$\sum_{j=1}^{N} P_{ij} \leq \frac{N}{2\gamma}(1 + o(1)).$$

The proof just uses the fact that for those vertices, $P_{ij}$ are typically negligible.

To get a lower bound on the spectral radius of $P$, we show that if one selects the submatrix $P_J := (P_{ij})_{i,j\in J}$ where $J$ is the collection of indices

$$J = \left\{1 \leq i \leq N, |X_i|^2 \leq 2\frac{\ln\gamma}{\gamma}\right\}, \tag{1}$$

the spectral radius of $P_J$ is almost $\frac{N}{2\gamma}$. This will give the desired estimate on the spectral radius of $P$.

**Lemma 5.** *Let $J$ be the subset defined in (1) and $P_J$ the associated sub matrix. Let $\mu_1(J)$ denote the largest eigenvalue of $P_J$. Then, with h.p., one has that*

$$\mu_1(J) \geq \frac{N}{2\gamma}(1 - o(1)).$$

The proof relies on the fact that vertices close to the origin get the most contribution to their connectivity from the other vertices close to the origin.

The constant $1/2$ that arises in the Proposition 2 is a direct consequence of the choice of the Gaussian kernel. Had we chosen a different kernel, this constant would have been different (once the width parameter $\gamma$ normalized appropriately). The techniques we developed can be used to compute it; this is merely a matter of computations, left as exercices.

## 2 A stochastic block model perturbed by a geometric graph

### 2.1 The model

We consider in this section the stochastic block model, with two communities (it can easily be extended to the coexistence of more communities), yet perturbed by a geometric graph. More precisely, we assume that each member $i$ of the network (regardless of its community) is characterized by an i.i.d. Gaussian vector $X_i$ in $\mathbb{R}^2$ with distribution $\mathcal{N}(0, I_2)$.

The perturbed stochastic block model is characterized by four parameters: the two probabilities of intra-inter connection of communities (denoted respectively by $p_1$ and $p_2 > 0$) and two connectivity parameters $\kappa, \gamma$, chosen so that $\max(p_1, p_2) + \kappa \leq 1$:

-In the usual stochastic block model, vertices $i$ and $j$ are connected with probability $r_{i,j}$ where

$$r_{ij} = \begin{cases} p_1 & \text{if } X_i, X_j \text{ belong to the same community} \\ p_2 & \text{otherwise} \end{cases},$$

where $p_1$ and $p_2$ are in the same order (the ratio $p_1/p_2$ is uniformly bounded).

-The geometric perturbation of the stochastic block model we consider is defined as follows. Conditionally on the values of $X_i$, the entries of the adjacency matrix $A = (A_{ij})$ are independent (up to symmetry) Bernoulli random variables with parameter $q_{ij} = \kappa e^{-\gamma|X_i - X_j|^2} + r_{ij}$.
We remind that the motivation is independent to incorporate the fact that members from two different communities can actually be "closer" in the latent space than members of the same community.
Thus in comparison with preceding model, the matrix $P$ of the geometric graph is now replaced with
$Q := \kappa P + \begin{pmatrix} p_1 J & p_2 J \\ p_2 J & p_1 J \end{pmatrix}$, where we assume, without loss of generality, that $X_i, i \leq N/2$ (resp. $i \geq N/2 + 1$) belong to the same community. The matrix

$$P_0 := \begin{pmatrix} p_1 J & p_2 J \\ p_2 J & p_1 J \end{pmatrix}$$

has two non zero eigenvalues which are $\lambda_1 = N(p_1 + p_2)/2$ with associated normalized eigenvector $v_1 = \frac{1}{\sqrt{N}}(1, 1, \ldots 1)^\top$ and $\lambda_2 = N(p_1 - p_2)/2$ associated to $v_2 = \sigma = \frac{1}{\sqrt{N}}(1, \ldots, 1, -1, \ldots -1)^\top$. Thus, in principle, communities can be detected from the eigenvectors of $P_0$ by using the fact that two vertices $i, j$ such that $v_2(i)v_2(j) = 1$ belong to the same community. Our method can be generalized (using sign vectors) to more complicated models where the two communities are of different size, as well as to the case of $k$ communities (and thus the matrix $P_0$ has $k$ non zero eigenvalues).

For the sake of notations, we write the adjacency matrix of the graph as :

$$A = P_0 + P_1 + A_c,$$

where $P_1 = \kappa P$ with $P$ the $N \times N$-random symmetric matrix with entries $(P_{ij})$ – studied in the previous section – and $A_c$ is, conditionnally on the $X_i$'s a random matrix with independent Bernoulli entries which are centered.

### 2.2 Separation of eigenvalues: the easy case

We are going to use spectral methods to identify communities. We therefore study in this section a regime where the eigenvalues of $A$ are well separated and the second eigenvector is approximately $v_2$, i.e. the vector which identifies precisely the two communities.

**Proposition 6.** *Assume that*

$$N(p_1 - p_2) \gg \sqrt{N} + \frac{N}{\gamma}.$$

*Then, with probability tending to 1, the two largest eigenvalues of $A$ denoted by $\rho_1 \geq \rho_2$ are given by*

$$\rho_i = \lambda_i(1 + o(1)), \ i = 1, 2.$$

*Furthermore, with probability tending to 1, associated normalized eigenvectors (with non negative first coordinate) denoted by $w_1$ and $w_2$ satisfy $\langle v_i, w_i \rangle = 1 - o(1); \ i = 1, 2.$*

Proposition 6 implies that, in the regime considered, the spectral analysis of the adjacency matrix can be directly used to detect communities, in the same way it is a standard technique for the classical stochastic block model (if $|p_1 - p_2|$ is big enough compared to $p_1 + p_2$, which is the case here). Finding the exact threshold $C_0$ such that if $N(p_1 - p_2) = C_0(\sqrt{N} + \frac{N}{\gamma})$ then the conclusion of Proposition 6 is still an open question.

## 2.3 Partial reconstruction when $\frac{N}{\gamma} \gg \sqrt{N(p_1 + p_2)}$

From Theorem 2.7 in [2], the spectral norm of $A_c$ cannot exceed

$$\rho(A_c) \leq \left( \sqrt{\kappa \frac{N}{\gamma}} + \sqrt{N(\frac{p_1 + p_2}{2} + \mathcal{O}(\frac{\kappa}{2\gamma}))} \right)(1 + \epsilon),$$

with probability tending to 1, since the maximal connectivity of a vertex does not exceed $N\left(\frac{p_1 + p_2}{2} + \frac{\kappa}{2\gamma}\right)(1 + o(1))$. In the specific regime where

$$\frac{\kappa N}{2\gamma} \ll \sqrt{N \frac{p_1 + p_2}{2}},$$

standard techniques [5] of communities detection would work, at the cost of additional perturbation arguments. As a consequence, we will concentrate on the reconstruction of communities when

$$\frac{\kappa N}{2\gamma} \gg \sqrt{N \frac{p_1 + p_2}{2}}.$$

This essentially means that the spectrum of $A_c$ is blurred into that of $P_1$. More precisely, we are from now going to consider the case where the noise induced by the latent random graph is of the same order of magnitude as the signal (which is the interesting regime):

$$\exists 0 < c, C < 1 \text{ s.t. } \lambda_2^{-1} \frac{\kappa N}{2\gamma} \in [c, C], \frac{\lambda_2}{\lambda_1} \in [c, C] \text{ and } \lambda_2 \gg \sqrt{\lambda_1}. \qquad (H_2)$$

If ($H_2$) holds, then the spectrum of $P_0 + P_1$ overwhelms that of $A_c$. As a consequence, the problem becomes that of community detection based on $P_0 + P_1$, which will be done using spectral methods.

To analyze the spectrum of $P_0 + P_1$, we will use extensively the resolvent identity [3] : consider $\theta \in \mathbb{C} \setminus \mathbb{R}$ and set $S = P_0 + P_1; R_S(\theta) = (S - \theta I)^{-1}, R_1(\theta) := (P_1 - \theta I)^{-1}$. One then has that

$$R_S(I + P_0 R_1) = R_1, \qquad (2)$$

where the variable $\theta$ is omitted for clarity when they are no possible confusion. Since $P_0$ is a rank two matrix, then $P_0$ can be written as $P_0 = \lambda_1 v_1 v_1^* + \lambda_2 v_2 v_2^*$ where $v_1$ and $v_2$ are the eigenvectors introduced before.

Eigenvalues of $S$ that are not eigenvalues of $P_1$ are roots of the rational equation $\det(I + P_0 R_1) = 0$:

$$\begin{aligned} \det(I + P_0 R_1) = \ & 1 + \lambda_1 \lambda_2 \langle R_1 v_1, v_1 \rangle \langle R_1 v_2, v_2 \rangle + \lambda_1 \langle R_1 v_1, v_1 \rangle \\ & + \lambda_2 \langle R_1 v_2, v_2 \rangle - \lambda_1 \lambda_2 \langle R_1 v_1, v_2 \rangle^2. \end{aligned} \qquad (3)$$

Let $\mu_1 \geq \mu_2 \geq \cdots \mu_N$ be the ordered eigenvalues of $P_1$ with associated normalized eigenvectors $w_1, w_2, \ldots, w_N$, then one has that $R_1(\theta) = \sum_{j=1}^{N} \frac{1}{\mu_j - \theta} w_j w_j^*$. Denote, for every $j \in \{1, .., N\}$, $r_j = \langle v_1, w_j \rangle$ and $s_j = \langle v_2, w_j \rangle$, so that Equation (3) rewrites into

$$\begin{aligned} \det(I + P_0 R_1(\theta)) =: f_{\lambda_1, \lambda_2}(\theta) = & 1 + \sum_{j=1}^{N} \frac{1}{\mu_j - \theta} (\lambda_1 r_j^2 + \lambda_2 s_j^2) \\ & + \lambda_1 \lambda_2 / 2 \sum_{j \neq k} \frac{1}{(\mu_j - \theta)(\mu_k - \theta)} (r_j s_k - r_k s_j)^2. \end{aligned} \qquad (4)$$

As mentioned before, we aim at using spectral methods to reconstruct communities based on the second eigenvector of $S$. As a consequence, these techniques may work only if (at least) two

eigenvalues of $S$, that are roots of $\det(I + P_0 R_1(\theta)) = 0$ exit the support of the spectrum of $P_1$, i.e., such that they are greater than $\mu_1$.

So we will examine conditions under which there exist two real solutions to Equation (4), with the restriction that they must be greater than $\mu_1$. If two such solutions exist, by considering the singularities in (2), then two eigenvalues of $S$ indeed lie outside the spectrum of $P_1$.

### 2.3.1 Separation of Eigenvalues in the rank two case.

We now prove that two eigenvalues of $S$ exit the support of the spectrum of $P_1$. Recall the definition of the function $f_{\lambda_1,\lambda_2}$ given in Equation (4) (or equivalently Equation (3)). One has that $\lim_{\theta\to\infty} f_{\lambda_1,\lambda_2}(\theta) = 1$, $f_{\lambda_1,\lambda_2}(\theta(\lambda_1)) < 0$ and similarly $f_{\lambda_1,\lambda_2}(\theta(\lambda_2)) < 0$, where $\theta(\cdot)$ is the function introduced in the rank 1 case. Thus two eigenvalues exit the spectrum of $P_1$ if

$$\lim_{\theta\to\mu_1^+} f_{\lambda_1,\lambda_2}(\theta) > 0.$$

First, let us make the following claim (a consequence of ($H_1$) and ($H_2$), see Lemma 9).

$$\liminf_{N\to\infty} \lambda_1 r_1^2 > 0. \qquad (H_3)$$

**Lemma 7.** *Assume ($H_1$), ($H_2$) and ($H_3$) hold and that there exists $\epsilon > 0$ such that*

$$\lambda_2 \geq 4\mu_1(1 + \epsilon) = 4\kappa \frac{N}{2\gamma}(1 + \epsilon).$$

*Then at least two eigenvalues of $P_0 + P_1$ separate from the spectrum of $P_1$.*

*Proof.* Let us first assume that

$\mu_1$ is isolated; there exists $\eta > 0$ such that for $N$ large enough $\mu_1 > \mu_2 + \eta$.

In this case, we look at the leading terms in the expansion of $g$ as $\theta$ approaches $\mu_1$. It holds that

$$f_{\lambda_1,\lambda_2}(\theta) \sim \frac{1}{\theta - \mu_1}\left(\lambda_1\lambda_2 \sum_{j\geq 2} \frac{1}{\theta - \mu_j}(r_1 s_j - r_j s_1)^2 - \lambda_1 r_1^2 - \lambda_2 s_1^2\right).$$

Using that the spectral radius of $P_1$ does not exceed $\mu_1$, we deduce that

$$f_{\lambda_1,\lambda_2}(\theta) \geq \frac{1}{\theta - \mu_1}\left(\frac{\lambda_1\lambda_2}{2\theta} \sum_{j\geq 2}(r_1 s_j - r_j s_1)^2 - \lambda_1 r_1^2 - \lambda_2 s_1^2\right)$$

$$\geq \frac{1}{\theta - \mu_1}\left(\frac{\lambda_1\lambda_2}{2\theta}(r_1^2 + s_1^2) - \lambda_1 r_1^2 - \lambda_2 s_1^2\right) \geq \frac{1}{\theta - \mu_1}\lambda_1(r_1^2 + s_1^2)\epsilon,$$

provided $\lambda_2 \geq 2\mu_1(1 + \epsilon)$. Note that if $\mu_1$ is isolated, the bound on $\lambda_2$ is improved by a factor of 2.

Now we examine the case where $\mu_1$ is not isolated. We then define

$$I^* := \{i :\ \limsup_{N\to\infty} \mu_i - \mu_1 = 0\},$$

and we define $\tilde{v}_i = \sum_{j\in I^*}\langle v_i, w_j\rangle w_j$, $i = 1, 2$. Then mimicking the above computations, we get

$$f_{\lambda_1,\lambda_2}(\theta) \geq \frac{1 + o(1)}{\theta - \mu_1}\left(\frac{\lambda_1\lambda_2}{4\theta}(||\tilde{v}_1^2|| + ||\tilde{v}_2^2||) - \lambda_1||\tilde{v}_1^2|| - \lambda_2||\tilde{v}_2^2||\right) \qquad (5)$$

so that two eigenvalues separate from the rest of the spectrum as soon as $\lambda_2 > 4\mu_1(1 + \epsilon)$. To get that statement we simply modify step by step the above arguments. This finishes the proof of Lemma 7 as soon as $\liminf_{N\to\infty} \lambda_1 r_1^2 > 0$. $\qquad \square$

The threshold exhibited for the critical value of $\lambda_2$ might not be the optimal one, however it is in the correct scale as we do not a priori expect a separation if $\lambda_2 \leq \mu_1$.

### 2.3.2 Partial reconstruction when $N\frac{p_1+p_2}{2}$ is known

In the specific case where $N\frac{p_1+p_2}{2}$ is known beforehand for some reason, it is possible to weakly recover communities using Davis-Kahan $\sin(\theta)$-theorem under the same condition than Lemma 7.

We recall that this theorem states that if $M = \alpha x x^\top$ and $\widetilde{M} = \beta \widetilde{x}\widetilde{x}^\top$ is the best rank-1 approximation of $M'$, where both $x$ and $\widetilde{x}$ are normalized to $\|x\| = \|\widetilde{x}\| = 1$, then

$$\min\left\{\|x - \widetilde{x}\|, \|x + \widetilde{x}\|\right\} \leq \frac{2\sqrt{2}}{\max\{|\alpha|, |\beta|\}}\|M - M'\|.$$

**Theorem 8.** *Assume that ($H_1$) and ($H_2$) hold and that there exists $\epsilon > 0$ such that*

$$\lambda_2 \geq 4\mu_1(1+\epsilon) \iff \frac{p_1 - p_2}{2} \geq \frac{2\kappa}{\gamma}(1+\epsilon),$$

*then weak recovery of the communities is possible.*

*Proof.* We are going to appeal to Davis-Kahan theorem with respect to

$$M = P_0 - N\frac{p_1 + p_2}{2}v_1v_1^\top = N\frac{p_1 - p_2}{2}v_2v_2^\top$$

and

$$M' = A - N\frac{p_1 + p_2}{2}v_1v_1^\top = P_0 + P_1 + A_c - N\frac{p_1 + p_2}{2}v_1v_1^\top = P_1 + A_c + M$$

As a consequence, let us denote by $\widetilde{x}$ the first eigenvector of $M'$ of norm 1 so that

$$\frac{1}{N}d_H(v_2, \text{sign}(\widetilde{x})) \leq \|v_2 - \widetilde{x}\|^2 \leq \frac{8}{\lambda_2^2}\|P_1 + A_c\|^2 = \frac{8}{\lambda_2^2}\mu_1^2(1 + o(1)).$$

Weak reconstruction is possible if the l.h.s. is strictly smaller than $1/2$, hence if $\lambda_2 \geq 4\mu_1(1+\varepsilon)$. $\square$

It is quite interesting that weak recovery is possible in the same regime where two eigenvalues of $P_0 + P_1$ separate from the spectrum of $P_1$. Yet the above computations imply that in order to compute $\widetilde{x}$, it is necessary to know $\frac{p_1+p_2}{2}$ (at least up to some negligible terms). In the standard stochastic block model, when $\kappa = 0$, this quantity can be efficiently estimated since the $\frac{N(N-1)}{2}$ edges are independently drawn with overall probability $\frac{p_1+p_2}{2}$. As a consequence, the average number of edges is a good estimate of $\frac{p_1+p_2}{2}$ up to its standard deviation. The latter is indeed negligible compared to $\frac{p_1+p_2}{2}$ as it is in the order of $\frac{1}{N}\sqrt{\frac{p_1+p_2}{2}}$.

On the other hand, when $\kappa \neq 0$, such trivial estimates are no longer available; indeed, we recall that the probability of having an edge between $X_i$ and $X_j$ is equal to $\frac{p_1+p_2}{2} + \frac{\kappa}{1+4\gamma}$, where all those terms are unknown (and moreover, activations of edges are no longer independent). We study in the following section, the case where $p_1 + p_2$ is not known. First, we will prove that Assumption ($H_3$) is actually always satisfied (notice that it was actually not required for weak recovery). In a second step, we will prove that *soft* recovery is possible, where we recall that this means we can output a vector $x \in \mathbb{R}^N$ such that $\|x\| = 1$ and $x^\top v_2$ does not converge to 0. Moreover, we also prove that weak (and exact) recovery is possible if the different parameters $p_1$, $p_2$ and $\frac{1}{\gamma}$ are sufficiently separated.

### 2.3.3 The case of unknown $p_1 + p_2$

We now proceed to show that Assumption ($H_3$) holds in the regime considered.

**Lemma 9.** *Under ($H_1$) and ($H_2$), one has that 1) for some constant $C > 0$, $\gamma r_1^2 \geq C$. and 2) for some $\epsilon > 0$ small enough, $\lambda_1 r_1^2 \geq \epsilon$.*

The first point of Lemma 9 implies ($H_3$) with an explicit rate if $\gamma \leq AN^{\frac{1}{2}}$ for some constant $A$. The second point proves this result in the general case.

**Theorem 10.** *If ($H_1$) and ($H_2$) hold true and $\lambda_1 > \lambda_2 + 2\frac{\kappa}{2\gamma}$ then the correlation $|w_2^\top v_2|$ is uniformly bounded away from 0 hence soft recovery is always possible. Moreover, if the ratio $\lambda_2/\mu_1$ goes to infinity, then $|w_2^\top v_2|$ tends to 1, which gives weak (and even exact at the limit) recovery.*

An (asymptotic) formula for the level of correlation is provided at the end of the proof.

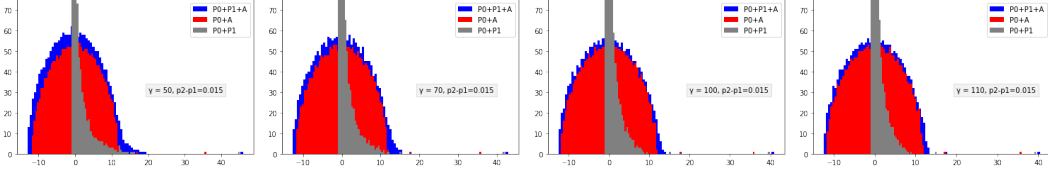

Figure 1: The spectrum of the different block models for different values of $\gamma$.

# 3 Experiments

The different results provided are theoretical and we proved that two eigenvalues separate from the bulk of the spectrum if the different parameters are big enough and sufficiently far from each other. And if they are too close to each other, it is also quite clear that spectral methods will not work. However, we highlight these statements in Figure 1. It illustrates the effect of perturbation on the spectrum of the stochastic block models for the following specific values: $N = 2000$, $p_1 = 2.5\%$, $p_2 = 1\%$, $\kappa = 0.97$ and $\gamma \in \{50, 70, 100, 110\}$. Notice that for those specific values with get $\lambda_1 = 35$, $\lambda_2 = 15$ and $\mu_1 \in \{20, 14.3, 10, 9.1\}$; in particular, two eigenvalues are well separated in the unperturbed stochastic block model.

The spectrum of the classical stochastic block model is coloured in red while the spectrum of the perturbed one is in blue ( the spectrum of the conditionnal adjacency matrix, given the $X_i$'s is in gray). As expected, for the value of $\gamma = 50$, the highest eigenvalue of $P_1$ is bigger than $\lambda_2$ and the spectrum of the expected adjacency matrix (in red) as some "tail". This prevents the separation of eigenvalues in the perturbed stochastic block model. Separation of eigenvalues starts to happen, empirically and for those range of parameters, around $\gamma = 70$ for which $\sqrt{\lambda_1} \leq \mu_1 = 10 \leq \lambda_2$.

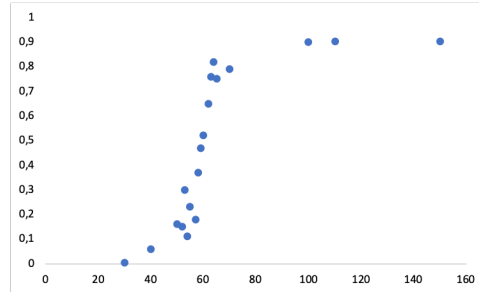

We also provide how the correlations between the second highest eigenvector and $\sigma$, the normalized vector indicating to which community vertices belong, evolve with respect to $\gamma$ for this choice of parameters, see Figure 2.

Figure 2: The correlation between the second highest eigenvector and the community vector goes from 0 to 0.9 around the critical value $\gamma = 60$.

# Conclusion

The method exposed hereabove can be generalized easily. In the case where there are $k \geq 2$ communities of different sizes, $P_0$ has rank $k$. If $k$ eigenvalues of $S$ exit the support of the spectrum of $P_1$, then communities may be reconstructed using a set of $k$ associated (sign) eigenvectors, whether the parameters are known or not.

We have proved that spectral methods to recover communities are robust to slight mis-specifications of the model, i.e., the presence of endogenous noise not assumed by the model (especially when $p_1 + p_2$ is not known in advance). Our results hold in the regime where $\frac{1}{\gamma} \gg \frac{\log N}{N}$ and with 2 communities (balancedness and the small dimension of latent variables were just assumed for the sake of computations) - those theoretical results are validated empirically by some simulations provided in the Appendix. Obtaining the same robustness results for more than 2 communities, for different types of perturbations and especially in the sparse regime $\frac{1}{\gamma} \sim p_i \sim \frac{1}{N}$ seems quite challenging as standard spectral techniques in this regime involve the non-backtracking matrix [5], and its concentration properties are quite challenging to establish.

## Broader Impact

This paper deals with theoretical detection of community in networks. Even if an entity wants to use community detection with some mercantile objectives (maybe in order to target some specific community), it would probably use spectral methods, no matter if the existing theory gives it guarantee that it is going to work. At worst, our paper will provide a positive answer: the very specific assumptions of stochastic block models are not required for theoretical (and certainly practical) recovery.

On the other hand, theoretical robustness results as ours can lead to substantial follow up research on finding the transition between regimes in complex models (almost ill-posed). Theory papers like this one are therefore win-win.

## Acknowledgments and Disclosure of Funding

This research was supported by the Institut Universitaire de France. It was also supported in part by a public grant as part of the Investissement d'avenir project, reference ANR-11-LABX-0056-LMH, LabEx LMH, in a joint call with Gaspard Monge Program for optimization, operations research and their interactions with data sciences and by the French Agence Nationale de la Recherche under the grant number ANR19-CE23-0026-04. No other competing interests.

## Footnotes

[1]We emphasize here that geometric interactions are defined through some kernel so that different recovery regimes can be identified with respect to a unique, simple width parameter $\gamma$. Similarly, the choice of the Gaussian kernel might seem a bit specific and arbitrary, but this purely for the sake of presentation: our approach can be generalized to other kernels (the "constants" will be different; they are defined w.r.t. the kernel chosen).

[2]The fact that $d = 2$ does not change much compared to $d > 3$; it is merely for the sake of computations; any Gaussian distribution $\mathcal{N}(0, \sigma^2 I_2)$ can be recovered by dividing $\gamma$ by $\sigma^2$.

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
