[Supplementary Material]

## A  Some illustrating experiments

The different results provided are theoretical and we proved that two eigenvalues separate from the bulk of the spectrum if the different parameters are big enough and sufficiently far from each other. And if they are too close to each other, it is also quite clear that spectral methods will not work. However, we highlight these statements in Figure 1.

Those figure illustrate the effect of perturbation on the spectrum of the stochastic block models for the following specific values: $N = 2000$, $p_1 = 2.5\%$, $p_2 = 1\%$, $\kappa = 0.97$ and $\gamma \in \{50, 70, 100, 110\}$. Notice that for those specific values with get $\lambda_1 = 35$, $\lambda_2 = 15$ and $\mu_1 \in \{20, 14.3, 10, 9.1\}$; in particular, two eigenvalues are well separated in the unperturbed stochastic block model.

The spectrum of the classical stochastic block model is colored in green while the spectrum of the perturbed one is in blue (in red is represented the spectrum of the conditionnal adjacency matrix, given the $X_i$'s). As expected, for the value of $\gamma = 50$, the highest eigenvalue of $P_1$ is bigger than $\lambda_2$ and the spectrum of the expected adjacency matrix (in red) as some "tail". This prevents the separation of eigenvalues in the perturbed stochastic block model.

Separation of eigenvalues starts to happen, empirically and for those range of parameters, around $\gamma = 70$ for which $\sqrt{\lambda_1} \leq \mu_1 = 10 \leq \lambda_2$.

We also provide how the correlations between the second highest eigenvector and $\sigma$, the normalized vector indicating to which community vertices belong, evolve with respect to $\gamma$ for this choice of parameters, see Figure 2.

## B  Additional results and technical proofs of Section 1

In this section, we gather additional results on the random graphs $P$, namely when it is connected (i.e., without isolated vertices) and whether it is possible to prove that some eigenvalues separate from the spectrum or not.

Then we will proceed to prove technical statements made in Section 1.

### B.1  The connectivity regime

Let us first consider a preliminary remark on the connectivity of the random graph. This result is for illustration purpose, as the connectivity (or not) of the geometric graphs would have no real impact on our main result, so we do not put too much emphasis on the exact threshold of connectivity. On the other hand, the result of Lemma 11 is rather intuitive as with very high probability, one of the $\|X_i\|^2$ are going to be of the order of $2\log(N)$, which indicate that the transition between connectivity or not should indeed be around $\log(N)/\log\log(N)$.

**Lemma 11.** *Assume that* $\frac{\log(N)}{\gamma \log\log N} \to \infty$ *as* $N \to \infty$. *Then one has that*

$$\mathbb{P}(\exists \text{ an isolated vertex } i, 1 \leq i \leq N) \to 0 \text{ as } N \to \infty.$$

*Proof.* Fix a vertex $i$. Conditionally on the $X_j$'s, the probability that $i$ is isolated is

$$\prod_{j \neq i}(1 - e^{-\gamma|X_i - X_j|^2}),$$

which we will integrate w.r.t. the distribution of independent $X_j$'s, $j \neq i$. Precisely, we get that the probability that there is an isolated vertex is upper-bounded by

$$\mathbb{E}\sum_i \prod_{j \neq i}(1 - e^{-\gamma|X_i - X_j|^2}) = N\mathbb{E}\Big(1 - \frac{1}{1 + 2\gamma}e^{-\frac{2\gamma}{1+2\gamma}\frac{|X_i|^2}{2}}\Big)^{N-1}$$

$$\leq N\Big(1 - \frac{1}{1 + 2\gamma}e^{-\frac{2\gamma}{1+2\gamma}\frac{A^2}{2}}\Big)^{N-1} + Ne^{\frac{-A^2}{2}}$$

Figure 1: The spectrum of the perturbed/unperturbed stochastic block models for $\gamma = 50$ (top left), 70 (top right), 100 (bottom left), 110 (bottom right).

Figure 2: The correlation between the second highest eigenvector and the community vector quickly grows from near 1 to 0.9 around the critical value $\gamma = 60$.

for every $A > 0$. In particular, the choice of $ne^{\frac{-A^2}{2}} = 1/\log(N)$ gives that the probability of having an isolated vertex is smaller than

$$N \exp\left(-\frac{N-1}{N\log(N)}\frac{1}{1+2\gamma}(N\log(N))^{\frac{1}{1+2\gamma}}\right) + \frac{1}{\log(N)}$$

So as soon as $\frac{\log(N)}{\gamma\log\log N} \to \infty$, the probability of having one isolated vertex goes to 0. $\qquad\square$

## B.2 Separation of eigenvalues

We now examine the possibility that some eigenvalues of $P$ separate from the rest of the spectrum, as it could interfere with standard spectral methods used in community detection. For that purpose, we are going to study the moments of the spectral measure of $P$.

**Proposition 12.** *Let $l \geq 2$ be a given integer, then the following holds:*

$$\lim_{N\to\infty}\frac{1}{\gamma}\mathbb{E}\mathrm{Tr}\left(\frac{2\gamma P}{N}\right)^l = \frac{1}{l^2}$$

$$\mathrm{Var}\frac{1}{\gamma}\mathrm{Tr}\left(\frac{\gamma P}{N}\right)^l = \mathcal{O}\left(\frac{1}{N}\right)$$

Proposition 12 implies in particular that the non-normalized spectral measure

$$\mu(P) = \sum_{i=1}^{N}\delta_{\mu_i}$$

has asymptotically some positive mass on large values in the order of $\frac{N}{\gamma}$. This does not prevent that the largest eigenvalue separates from the others but it does not hold that the largest eigenvalue computed in Proposition 2 overwhelms the remaining eigenvalues.

Proposition 12 roughly states that the largest eigenvalue does not macroscopically separate from the rest of the spectrum. Instead it is blurred into a cloud of large eigenvalues and thus cannot be distinguished. Notice that this phenomenon is rather different from the standard stochastic block model for which there exists a regime (in the average degree of the graph) where a finite number of eigenvalues really overwhelm the rest of the spectrum.

*Proof.* We use the fact that the $X_i$'s are Gaussian random variables to give an explicit formula for the moments of the spectral measure $\mu(P)$. Let us use the standard method to derive its moments: let $l > 1$ be given. One has that

$$\mathbb{E}\sum_{i=1}^{N}\mu_i^l = \mathbb{E}\mathrm{Tr}P^l = \sum_{i_1,i_2,\ldots,i_l}\mathbb{E}\prod_{j=1}^{l}P_{i_j i_{j+1}}, \tag{6}$$

using the convention that $i_{l+1} = i_1$. Note that there may be some coincidences among the vertices $i_1, i_2, \ldots, i_l$ chosen in $\{1, \ldots, N\}$. We forget for a while the precise labels of these vertices and denote them by $w_1, w_2, \ldots, w_l$ instead (keeping track of the coincidences however).

For each possible choice of the set of coincidences in (6), we denote by $k \geq 1$ the number of pairwise distinct indices (that we again label $w_1, w_2, \ldots w_k$). We associate a graph $G_k$ on the vertices $\{w_1, w_2, \ldots w_k\}$ by simply drawing the edges $(w_j, w_{j+1}), j = 1, \ldots, l$. Note that the graph may have multiple edges. It has no loops because $P_{ii} = 0$, for any vertex $i$. Let $C_l$ denote the simple cycle with vertices $1, 2, \ldots, l$ in order. Then this graph corresponds to the case where there is no coincidence. When there are some coincidences, some vertices from $C_l$ are pairwise identified (excluding the possibility that subsequent vertices along the cycle are identified due to the fact that loops are not allowed). For $k < l$ we denote by $\mathbf{G_k}$ the set of such graphs obtained by pairwise identifications of vertices from $C_l$ (excluding subsequent vertices). Note that $\mathbf{G_l} = \{C_l\}$.

Then one has that

$$\mathbb{E} \sum_{i=1}^{N} \mu_i^l = \sum_{k=2}^{l} \sum_{G_k \in \mathbf{G_k}} N(N-1)\cdots(N-k+1)\mathbb{E} \prod_{e \in G_k} P_e, \qquad (7)$$

where in the above formula we have chosen the set of actual vertices among $\{1, \ldots, N\}$ and each edge $e \in G_k$ is repeated with its multiplicity in the product. By standard Gaussian integration, using that $P_{(ij)} = \exp\{-\gamma\|X_i - X_j\|^2\}$, one can easily check that

$$\mathbb{E} \prod_{e \in G_k} P_e = \left(\det(I + 2\gamma L_{G_k})\right)^{-1}, \qquad (8)$$

where $L_{G_k}$ is the Laplacian of $G_k$: we recall that the Laplacian of a graph $G = (V, E)$, $V = \{1, \ldots, k\}$ is the $k \times k$ matrix whose entries are

$$L_{ii} = -\deg(i), i = 1, 2, \ldots, k; L_{ij} = m_{ij}, i < j,$$

where $m_{ij}$ is the multiplicity of the non oriented edge $(i, j)$.

We now perform the expansion of $\det(I + 2\gamma L_{G_k})$ according to the powers of $\gamma$. By the matrix tree theorem (see [6] e.g.), one has that

$$\det(I + 2\gamma L_{G_k}) = (2\gamma)^{k-1} k \times \sharp\{\text{spanning trees of } G_k\} + \sum_{i=2}^{k} (2\gamma)^{k-i} a_{k,i}, \qquad (9)$$

for some coefficients $a_{k,i}$ which can be easily deduced from some minors of $L_{G_k}$. Combining now equations (7), (8), (9), and using that $C_l$ has $l$ spanning trees, we deduce that

$$
\begin{aligned}
\mathbb{E} \sum_{i=1}^{N} \mu_i^l \quad &= N^l(1 + o(1)) \frac{1}{(2\gamma)^{l-1} l^2 (1 + o(\gamma^{-1})} \\
&\quad + \sum_{k=2}^{l-1} N^k (1 + o(k^2/N)) \frac{1}{(2\gamma)^{k-1} c_k (1 + o(\gamma^{-1})} \\
&= \frac{N^l}{(2\gamma)^{l-1} l^2} \left(1 + \mathcal{O}(\gamma^{-1}) + \mathcal{O}\left(\frac{\gamma}{N}\right)\right). \qquad (10)
\end{aligned}
$$

In the second line of (10), the constant $c_k$ is given by

$$c_k^{-1} = \sum_{G_k \in \mathbf{G_k}} \frac{1}{k \sharp\{\text{spanning trees of } G_k\}}.$$

Thus we have proved the first statement of Proposition 12.

Let us now turn to the variance :

$$\text{Var}(\text{Tr}P^l) = \mathbb{E}\left(\text{Tr}P^l \text{Tr}P^l\right) - \left(\mathbb{E}\text{Tr}P^l\right)^2.$$

We again developp the product

$$\mathrm{Tr}P^l\mathrm{Tr}P^l = \sum_{i_1,i_2,\ldots,i_l} \prod_{k=1}^{l} P_{i_j i_{j+1}} \sum_{i'_1,i'_2,\ldots,i'_l} \prod_{k=1}^{l} P_{i'_j i'_{j+1}}$$

and draw the associated graphs (forgetting the labels) on possibly $2l$ vertices. If the two graphs are disconnected (this means that the two sets $\{i_1, i_2, \ldots, i_l\}$ and $\{i'_1, i'_2, \ldots, i'_l\}$ are disjoint, then the expectation of the product splits by independance. The combined contribution of each subgraph to the variance will thus be in the order of $l^2/N$ times $(\mathbb{E}\mathrm{Tr}P^l)^2$. This comes from the fact that one has to choose $2k$ pairwise distinct indices when combining the two graphs (while twice $k$ pairwise distinct indices when considering the squared expectation of the Trace). Thus, by definition of the variance, the only graphs which are contributing to the variance are those for which at least one vertex from $\{i_1, i_2, \ldots, i_l\}$ and $\{i'_1, i'_2, \ldots, i'_l\}$ coincide. This means that using the same procedure as above, one can restrict to the set of graphs $G_k, k \leq 2l - 1$ which are obtained from $C_{2l}$ by at least one identification.

From the above it is not difficult to check that $\mathrm{Var}\frac{1}{\gamma}\mathrm{Tr}\left(\frac{\gamma P}{N}\right)^l = \mathcal{O}\left(\frac{1}{N}\right)$. This finishes the proof of Proposition 12. □

### B.3 Proof of Proposition 1

We first show that there exists a constant $C_1$ such that

$$\frac{\mu_1(P)}{NC_1(\gamma_0)} \geq 1$$

for $N$ large enough. For $i = 1, \ldots, N$ we set $d(i) := \sum_j P_{ij}$, which we call "the degree" of $i$. By the Perron Frobenius theorem the largest eigenvalue of $P$ cannot exceed the maximal degree of a vertex, (which can be proved to be strictly greater than $\frac{N}{1+4\gamma}$). However the number of vertices whose degree is such high is negligible with respect to $N$ (it is not obvious such a number grows to infinity actually). Because all the entries of $P$ are positive, one knows that the largest eigenvalue of $P$ is simple and is equal to the spectral radius of $P$. Furthermore, one has that

$$\mu_1(P) = \lim_{l \to \infty} \frac{\langle v_1, P^l v_1 \rangle}{\langle v_1, P^{l-1} v_1 \rangle}$$

where $\sqrt{N}v_1 = \tilde{v}_1 = (1, 1, \ldots, 1)^t$. Actually we are going to show that

$$\mu_1(P)^2 = (1 + o(1))\frac{\langle v_1, P^{2l+2} v_1 \rangle}{\langle v_1, P^{2l} v_1 \rangle} \text{ for } l = \ln N.$$

First one has that

$$\mu_1(P)^2 \geq \frac{\langle v_1, P^{2l+2} v_1 \rangle}{\langle v_1, P^{2l} v_1 \rangle} \text{for } l = \ln N.$$

Now we show some concentration estimates for both the numerator and denominator, for $l \sim \ln N$ showing that to the leading order they concentrate around their mean which is enough to show that

$$\mu_1 \geq C_1(\gamma)N(1 + o(1)).$$

Observe that $\langle \tilde{v}_1, P^l \tilde{v}_1 \rangle = \sum_{i,j,i_1,\ldots,i_{l-1}} P_{ii_1} P_{i_1 i_2} P_{i_{l-1}j}$ is a sum of at most $N^{l+1}$ terms. Each of the summands if a function of the Gaussian vector $X = (X_1, X_2, \ldots, X_N)^t$. We are going to show that $X \mapsto \sum_{i,j,i_1,\ldots,i_{l-1}} P_{ii_1} P_{i_1 i_2} P_{i_{l-1}j}$ is Lipschitz with Lipschitz constant in the order of $N^{(2l+1)/2}$ for some constant $C$ large enough. As $\mathbb{E}\sum_{i,j,i_1,\ldots,i_{l-1}} P_{ii_1} P_{i_1 i_2} P_{i_{l-1}j} = (NC_2(\gamma_0))^{l+1}(1 + o(1))$ for some constant $C_2(\gamma_0) > 0$, this will be enough to ensure using standard concentration arguments for Gaussian vectors that

$$\mathbb{P}\left(|\sum_{i,j,i_1,\ldots,i_{l-1}} P_{ii_1} P_{i_1 i_2} P_{i_{l-1}j} - (C_2(\gamma_0)N)^{l+1}| \geq AN^{(2l+1)/2}\right) \leq 2e^{-2A^2}.$$

Thus this implies that a.s.

$$\lim_{N\to\infty} \frac{\sum_{i,j,i_1,\dots,i_{l-1}} P_{ii_1} P_{i_1 i_2} P_{i_{l-1} j}}{(C_2(\gamma_0)N)^{l+1}} = 1.$$

Consider two vectors $X$ and $Y$. One has that

$$\left| \sum_{i,j,i_1,\dots,i_{l-1}} P_{ii_1} P_{i_1 i_2} P_{i_{l-1}j}(X) - \sum_{i,j,i_1,\dots,i_{l-1}} P_{ii_1} P_{i_1 i_2} P_{i_{l-1}j}(Y) \right|$$

$$\leq \sum_{k=0}^{l-1} \sum_{i,j,i_1,\dots,i_{l-1}} P_{ii_1}(X) P_{i_1 i_2}(X) \left| P_{i_k i_{k+1}}(X) - P_{i_k i_{k+1}}(Y) \right| P_{i_{k+1} i_{k+2}}(Y) \dots P_{i_{l-1}j}(Y)$$

$$\leq \alpha \sum_{k=0}^{l-1} \sum_{i,j,i_1,\dots,i_{l-1}} \prod_{l=0}^{k-1} P_{i_l i_{l+1}}(X) \prod_{l=k+1}^{l-1} P_{i_l i_{l+1}}(Y) \Big| |X_{i_k} - X_{i_{k+1}}| - |Y_{i_k} - Y_{i_{k+1}}| \Big|, \tag{11}$$

where in the last line we have used the fact that $x \mapsto e^{-\gamma x^2}$ is $\alpha$-Lipschitz. The constant $\alpha$ can be chosen as $\alpha = 4\sqrt{\gamma} \sup_x |xe^{-x^2}|$. Consider the sum in (11). We note $\sum_*$ the sum over indices $i, j, i_1, \dots, i_{l-1}$ and $k$ in the following. One has that

$$\sum_* \prod_{l=0}^{k-1} P_{i_l i_{l+1}}(X) \prod_{l=k+1}^{l-1} P_{i_l i_{l+1}}(Y) \Big| |X_{i_k} - X_{i_{k+1}}| - |Y_{i_k} - Y_{i_{k+1}}| \Big|$$

$$\leq \sqrt{\sum_* \prod_{l=0}^{k-1} P_{i_l i_{l+1}}^2(X) \prod_{l=k+1}^{l-1} P_{i_l i_{l+1}}^2(Y)} \sqrt{\sum_* \Big| |X_{i_k} - X_{i_{k+1}}| - |Y_{i_k} - Y_{i_{k+1}}| \Big|^2}$$

$$\leq N^{\frac{l+1}{2}} N^{\frac{l-1}{2}} \left( \sum_k 8|X - X_k v_1 - (Y - Y_k v_1)|^2 \right)^{\frac{1}{2}}$$

$$\leq C N^{(2l+1)/2} ||X - Y||.$$

We now show that

$$\mu_1(P)^2 \leq (1 + o(1)) \frac{\langle v_1, P^{2l+2} v_1 \rangle}{\langle v_1, P^{2l} v_1 \rangle} \text{ for } l = \ln N.$$

Denote by $w_i, i = 1, \dots, N$ a set of orthonormalized eigenvectors of $P$. Equivalently the above means that

$$\sum_{i>1} \mu_i^{2l}(\mu_1^2 - \mu_i^2)\langle w_i, v_1 \rangle^2 = o(1) \sum_{i\geq 1} \mu_i^{2l+2}\langle w_i, v_1 \rangle^2.$$

Fix $\epsilon > 0$. Set $r^2 := \sum_{i:\mu_1 - |\mu_i| < \epsilon} \langle w_i, v_1 \rangle^2$. The first sum in the above then does not exceed:

$$2\epsilon r^2 \mu_1^{2l+1} + \mu_1^{2l+2}(1 - r^2)(1 - \epsilon)^{2l}.$$

This is $o(1)\mu_1^{2l+2}r^2$ provided that $r^2 \geq \eta$ for some $\eta > 0$. This is the fact we prove below. To that aim we show that $\langle w_1, v_1 \rangle^2 \geq \eta$. Using that $w_1$ (associated to $\mu_1$) has non negative coordinates and is normalized to 1, one has that $\langle w_1, v_1 \rangle \geq \frac{1}{\sqrt{N}|w_1|_\infty}$. Thus it is enough to show that $\limsup \sqrt{N}|w_1|_\infty < \infty$. Assume this is not the case : then there exists a sequence $A_N \to \infty$ such that $\sqrt{N}|w_1|_\infty \geq A_N$ (along some subsequence). In particular let $w_{i_0} = \max w_i \geq \frac{A_N}{\sqrt{N}}$. Fix $\delta > 0$ small. Set $J := \{j, w_j \geq \delta w_{i_0}\}$. Then one has that $\sharp J \leq \frac{N}{\delta^2 A_N^2} \ll N$. Using this in the expression

$$\mu_1 = \sum_{j\in J} P_{i_0 j} \frac{w_j}{w_{i_0}} + \sum_{j\notin J} P_{i_0 j} \frac{w_j}{w_{i_0}}$$

one deduces that

$$\mu_1 \leq N\delta + \sharp J,$$

which is a contradiction. This finishes the proof of Proposition 1.

## B.4 Proof of Lemma 3

Let us first introduce some notations and key results for the proof. The function

$$\theta : r \geq 0 \mapsto \theta(i, r) := \int_{D(X_i, \sqrt{r})} \frac{1}{2\pi} e^{-|x|^2/2} d\lambda_2(x),$$

 where $D(X_i, \sqrt{r})$ is the disk centered at $X_i$ of radius $\sqrt{r}$.

Notice that the following holds for all $r > 0$

$$e^{-\frac{\|X_i\|^2}{2}} \left(1 - e^{-\frac{r}{2}}\right) e^{-2\|X_i\|\sqrt{r}} \leq \theta(i, r) \leq e^{-\frac{\|X_i\|^2}{2}} \left(1 - e^{-\frac{r}{2}}\right) e^{2\|X_i\|\sqrt{r}}.$$

It also holds that

$$2e^{-\|X_i\|^2}(1 - e^{-r}) \leq \theta(i, r) \leq e^{-\frac{\|X_i\|^2}{4}} (e^{\frac{r}{2}} - 1)$$

and moreover if $r_1 > r_0$ then we immediately have

$$\theta(i, r_1) - \theta(i, r_0) \leq \frac{r_1 - r_0}{2}.$$

Conditionally on $X_i$, the number of vectors among the $X'_j$s whose distance to $X_i$ falls in the interval $I$ is a binomial random variable $\text{Bin}(N - 1, \theta(i, l(I)))$. So we recall the following basic concentration argument (see equivalently Theorem 2.6.2 in [21]). Let $Z$ be a binomial random variable with distribution $\text{Bin}(m, p)$. There exists a constant $\alpha > 0$ ( if $p < 4/5$, one can choose $\alpha = 1/32$) such that for any $C > 0$, one has

$$\mathbb{P}\left(|Z - mp| \geq C\sqrt{mp}\right) \leq 2e^{-\alpha C^2}.$$

We can now turn to the proof of Lemma 3 itself. Let $\varepsilon > 0$ be fixed (its specific value is tuned at the end of the proof) and $i \in [N]$ be a fixed index such that $|X_i|^2 \leq \frac{2\ln \gamma}{\gamma}$. We are going to show that

$$S := \sum_{j=1}^{N} e^{-\gamma|X_i - X_j|^2} = c_0 \frac{N}{\gamma} \left(1 \pm o(1)\right),$$

where

$$c_0 := \lim_{N \to \infty} \frac{\gamma}{N} \sum_{k=1}^{2\frac{\ln \gamma}{\varepsilon}} \mathbf{n}_{\mathbf{k}}^{(\mathbf{i})} e^{-k\varepsilon},$$

with $\forall k = 1, \ldots, 2\frac{\ln \gamma}{\varepsilon}$,

$$\mathbf{n}_{\mathbf{k}}^{(\mathbf{i})} := N\left(\theta\left(i, \frac{(k+1)\varepsilon}{\gamma}\right) - \theta\left(i, \frac{k\varepsilon}{\gamma}\right)\right).$$

As $\gamma$ goes to infinity with $N$, then it holds that

$$Ne^{-\frac{\|X_i\|^2}{2}} \left(\frac{\varepsilon}{2\gamma} - \mathcal{O}(\frac{\ln^2 \gamma}{\gamma^2})\right) \leq \mathbf{n}_{\mathbf{k}}^{(\mathbf{i})} \leq N\frac{\varepsilon}{2\gamma}$$

 so that if $\|X_i\|^2 \leq 2\frac{\ln \gamma}{\gamma}$ then $\mathbf{n}_{\mathbf{k}}^{(\mathbf{i})} \simeq \frac{N\varepsilon}{2\gamma}$ which ensures that $c_0 = \frac{1}{2}(1 + o(1))$ is well-defined.

 To control $S$, we split this sum into three parts, depending on the distances from $X_j$ to $X_i$, as follows

$$S = \underbrace{\sum_{j:d^2(X_i, X_j) \in [\frac{\varepsilon}{\gamma}, \frac{2\ln \gamma}{\gamma}]} e^{-\gamma|X_i - X_j|^2}}_{S_1} + \underbrace{\sum_{j:d^2(X_i, X_j) < \frac{\varepsilon}{\gamma}} e^{-\gamma|X_i - X_j|^2}}_{S_2}$$

$$+ \underbrace{\sum_{j:d^2(X_i, X_j) > \frac{2\ln \gamma}{\gamma}} e^{-\gamma|X_i - X_j|^2}}_{S_3}$$

We first focus on $S_1$ that we are going to further decompose as a function of the distance from $X_j$ to $X_i$ : define for $k \in \{1, \ldots, 2\frac{\ln \gamma}{\varepsilon}\}$

$$n_k^{(i)} := \sharp\left\{l, d^2(X_l, X_i) \in \left[\frac{k\varepsilon}{\gamma}, \frac{(k+1)\varepsilon}{\gamma}\right[\right\}.$$

Then one has

$$
\begin{aligned}
S_1 &\leq \sum_{k=1}^{2\frac{\ln \gamma}{\varepsilon}} e^{-k\varepsilon} n_k^{(i)} \\
&= \sum_{k=1}^{2\frac{\ln \gamma}{\varepsilon}} e^{-k\varepsilon} \mathbf{n_k^{(i)}} + \sum_{k=1}^{2\frac{\ln \gamma}{\varepsilon}} e^{-k\varepsilon} \left(n_k^{(i)} - \mathbf{n_k^{(i)}}\right) \\
&= \frac{N}{2\gamma}\left(1 + o(1)\right) + \sum_{k=1}^{2\frac{\ln \gamma}{\varepsilon}} e^{-k\varepsilon}(n_k^{(i)} - \mathbf{n_k^{(i)}}),
\end{aligned}
$$

where the last equality comes from the approximation of $\mathbf{n_k^{(i)}}$ as $N$ and $\gamma$ goes to infinity. It also holds that

$$
\begin{aligned}
S_1 &\geq \sum_{k=1}^{2\frac{\ln \gamma}{\varepsilon}} e^{-(k+1)\varepsilon} n_k^{(i)} \\
&= \sum_{k=1}^{2\frac{\ln \gamma}{\varepsilon}} e^{-(k+1)\varepsilon} \mathbf{n_k^{(i)}} + \sum_{k=1}^{2\frac{\ln \gamma}{\varepsilon}} e^{-(k+1)\varepsilon} \left(n_k^{(i)} - \mathbf{n_k^{(i)}}\right) \\
&\geq \frac{N}{2\gamma}\left(1 - 2\varepsilon - o(1)\right) + \sum_{k=1}^{2\frac{\ln \gamma}{\varepsilon}} e^{-(k+1)\varepsilon}(n_k^{(i)} - \mathbf{n_k^{(i)}}).
\end{aligned}
$$

It remains to control the different errors $n_k^{(i)} - \mathbf{n_k^{(i)}}$. It holds that,

$$\mathbb{P}_{X_i}\left(\exists 1 \leq k \leq \frac{2\ln \gamma}{\varepsilon}, \; |n_k^{(i)} - \mathbf{n_k^{(i)}}| \geq \varepsilon \mathbf{n_k^{(i)}}\right) \leq 8\frac{\ln(\gamma)}{\varepsilon} e^{-\alpha\varepsilon^2 \frac{N}{4\gamma}},$$

because each $\mathbf{n_k^{(i)}} \simeq \frac{N\varepsilon}{2\gamma}$ as $\gamma$ increase to infinity with $N$. At the end, we obtained that for each $X_i$ such that $\|X_i\|^2 \leq \frac{2\log(\gamma)}{\gamma}$, then

$$\left|S_1 - \frac{N}{2\gamma}\right| \leq \frac{N}{2\gamma}(3\varepsilon + o(1)) \quad \text{with proba at least } 1 - 8\frac{\ln(\gamma)}{\varepsilon} e^{-\alpha\varepsilon^3 \frac{N}{4\gamma}}. \tag{12}$$

Let us now focus on $S_2$ which is obviously smaller than $n_0^{(i)}$ where

$$n_0^{(i)} := \sharp\{j, d^2(X_i, X_j) < \frac{\varepsilon}{\gamma}\}.$$

Moreover, because of the concentration of binomials, it holds that

$$\mathbb{P}_{X_i}\left(n_0^{(i)} \geq 2N\theta(i, \frac{\varepsilon}{\gamma})\right) \leq 2e^{-\alpha N\theta(i, \frac{\varepsilon}{\gamma})}.$$

Now as $\gamma$ goes to infinity with $N$, then for $\gamma$ large enough, the following holds

$$\frac{\varepsilon}{4\gamma} \leq \theta(i, \frac{\varepsilon}{\gamma}) \leq \frac{\varepsilon}{2\gamma}$$

which ensures that

$$\mathbb{P}_{X_i}\left(n_0^{(i)} \geq \frac{N\varepsilon}{\gamma}\right) \leq 2e^{-\alpha \frac{N\varepsilon}{4\gamma}}.$$

390 As a consequence we have shown that

$$S_2 \leq \frac{N\varepsilon}{\gamma} \quad \text{with probability at least } 1 - 2e^{-\frac{\alpha}{4}\frac{N\varepsilon}{\gamma}}. \tag{13}$$

391 Last, by the very definition of $S_3$, it always holds that

$$S_3 \leq Ne^{-\ln\gamma^2} \leq \frac{N}{\gamma^2}. \tag{14}$$

Combining (12), (13) and (14), we obtain that with probability at most

$$2e^{-\frac{\alpha}{4}\frac{N\varepsilon}{\gamma}} + 8\frac{\ln(\gamma)}{\varepsilon}e^{-\alpha\varepsilon^3\frac{N}{4\gamma}}$$

one has that

$$\left| S - \frac{N}{2\gamma} \right| \leq \frac{N}{2\gamma}\big(5\varepsilon + o(1)\big).$$

392 As a consequence, as $N$ grows to infinity, one has

$$\mathbb{P}\left( \exists i : |X_i|^2 \leq \frac{2\ln\gamma}{\gamma} \text{ and } \left| \sum_{j=1}^{N} e^{-\gamma|X_i - X_j|^2} - \frac{N}{2\gamma} \right| \geq \frac{N}{2\gamma}\big(5\varepsilon + o(1)\big) \right)$$

$$\leq 4N\frac{\ln\gamma}{\gamma}\left( e^{-\frac{\alpha}{4}\frac{N\varepsilon}{\gamma}} + 4\frac{\ln(\gamma)}{\varepsilon}e^{-\alpha\varepsilon^3\frac{N}{4\gamma}} \right) \to 0$$

393 by choosing $\varepsilon = \left( \frac{N}{\gamma\ln\gamma} \right)^{-1/4}$ (so that $\varepsilon$ goes to 0 as intended) and because $\frac{N}{\gamma\ln\gamma}$ goes to infinity.
394 This proves Lemma 3.

## B.5 Proof of Lemma 4

396 The proof is almost identical to that of Lemma 3. The only difference is that we cannot approximate
397 $\mathbf{n_k^{(i)}}$ by $\frac{N\varepsilon}{2\gamma}$ because $e^{-\frac{\|X_i\|^2}{2}}$ might go to 0. Yet it still holds that $\mathbf{n_k^{(i)}} \leq \frac{N\varepsilon}{2\gamma}$. And thus, we can easily
398 prove the weaker statement

$$\mathbb{P}\left( \exists i, \sum_{j=1}^{N} e^{-\gamma|X_i - X_j|^2} \geq \frac{N}{2\gamma}\big(1 + 5\varepsilon + o(1)\big) \right)$$

$$\leq 4N\left( e^{-\frac{\alpha}{4}\frac{N\varepsilon}{\gamma}} + 4\frac{\ln(\gamma)}{\varepsilon}e^{-\alpha\varepsilon^3\frac{N}{4\gamma}} \right) \to 0$$

399 with the same choice of $\varepsilon$, assuming Assumption $(H_1)$ holds.

## B.6 Proof of Lemma 5

401 If we can show that for any $i \in J$ and with probability close to 1, it holds that

$$\sum_{j\notin J} P_{ij} \ll \frac{N}{\gamma}, \tag{15}$$

402 then the result would be a direct consequence of Lemma 3.

By the very definition of $J$, if $j \notin J$, then necessarily $|X_j|^2 \geq \frac{2\ln\gamma}{\gamma}$. Notice that $|X_j|^2 \geq (3+\epsilon)\frac{\ln\gamma}{\gamma}$
then $\gamma|X_i - X_j|^2 \geq (1+\epsilon)\ln\gamma$ so that for any $i \in J$, this immediately yields that

$$\sum_{j,|X_j|^2 \geq (3+\epsilon)\frac{\ln\gamma}{\gamma}} P_{ij} \leq \frac{N}{\gamma^{1+\epsilon}} \ll \frac{N}{\gamma}.$$

This is enough to obtain (15) for the contribution of such indices. Note also that the same argument is
valid to get (15) for the subsum (keeping $i \in J$ fixed)

$$\sum_{j,\gamma|X_i - X_j|^2 \geq (1+\epsilon)\ln\gamma} P_{ij} \leq \frac{N}{\gamma^{1+\epsilon}} \ll \frac{N}{\gamma}.$$

Thus we only need to consider indices $i \in J$ and $j \notin J$ such that $\gamma|X_i - X_j|^2 \leq (1 + \epsilon) \ln \gamma$. This implies in particular that necessarily $\|X_j\|^2 \leq 8\frac{\ln(\gamma)}{\gamma}$. Consider therefore such an index $i$ and let

$$S := \sum_{j:\gamma|X_j - X_i|^2 \leq (1+\epsilon) \ln \gamma, |X_j|^2 \geq \frac{2 \ln \gamma}{\gamma}} e^{-\gamma|X_i - X_j|^2}.$$

Because $\|X_j\|^2 \leq 8\frac{\ln \gamma}{\gamma}$ then the number of indices $j \notin J$ is smaller than than $16N\frac{\ln \gamma}{\gamma}$ with probability at least $1 - e^{-\alpha 8N\frac{\ln \gamma}{\gamma}}$. As a consequence, the sum above is composed of at most $16N\frac{\ln \gamma}{\gamma}$ terms, all smaller than 1. Obviously, if they are all smaller than $\frac{1}{\ln^2 \gamma}$ then $S \leq 16\frac{N}{\gamma \ln \gamma} \ll \frac{N}{\gamma}$.

So this implies that it only remains to control the sum $S$ for indices $i \in J$ such that for some $j \notin J$ it holds that $\|X_i - X_j\|^2 \leq \frac{4 \ln \ln \gamma}{\gamma}$. This implies that such indices $i \in J$ must satisfy

$$2\frac{\ln \gamma}{\gamma} \geq |X_i|^2 \geq 2\frac{\ln \gamma}{\gamma} \left(1 - 2\sqrt{2\frac{\ln \ln \gamma}{\ln \gamma}}\right).$$

And, using the same argument as before, there are at most $8\frac{N}{\gamma}\sqrt{\ln \gamma \ln \ln \gamma}$ such indices with arbitrarily high probability (as $\gamma$ goes to infinity). On the other hand, $\sharp J$ (the cardinality of $J$) is, with arbitrarily high probability, of the order of $N\frac{\ln \gamma}{\gamma}$

This gives a lower bound on the spectral radius of $P_J$: let $v$ be the unit vector $v = \frac{1}{\sqrt{\sharp J}}(1, \ldots, 1)^t$ (of dimension $\sharp J$). Then

$$\langle P_J v, v \rangle \geq \frac{\sharp J - 8\frac{N}{\gamma}\sqrt{\ln \gamma \ln \ln \gamma}}{\sharp J} \frac{N}{2\gamma}(1 - o(1))$$

$$\geq \frac{N}{2\gamma}(1 - o(1))\left(1 - 8\sqrt{\frac{\ln \ln \gamma}{\ln \gamma}}\right)$$

$$\geq \frac{N}{2\gamma}(1 - o(1)).$$

Hence Lemma 5 is proved.

## C   Technical proofs of Section 2

### C.1   Proof of Proposition 6

The preceding proof can be easily modified to obtain the following bounds on the spectral radii : there exist constants $c_0 = 1/2, C > 0$ so that with high probability

$$\rho(P_1) \leq c_0 \frac{N}{\gamma}; \; \rho(A_c) \leq C\sqrt{N}.$$

Following [11], we first prove that the largest eigenvalue of $A$ is up to a negligible error (in the appropriate regime of $p_1, p_2, \gamma$) that of $P_0$. More precisely, it holds with arbitrarily high probability that

$$\langle Av_1, v_1 \rangle = N\frac{p_1 + p_2}{2} + \mathcal{O}\left(\frac{N}{\gamma} + \sqrt{N\left(\frac{p_1 + p_2}{2} + \frac{\kappa}{2\gamma}\right)}\right).$$

It easily follows that the largest eigenvalue $\rho_1(A)$ of $A$ satisfies

$$\rho_1 \geq \lambda_1\left(1 + \mathcal{O}(\frac{1}{\gamma(p_1 + p_2)} + \frac{1}{\sqrt{N}})\right).$$

In addition decomposing a normalized eigenvector $v$ associated to $\rho_1$ as

$$v = r_1 v_1 + r_2 v_2 + \sqrt{1 - r^2}w$$

for some normalized vector $w$ orthogonal to $v_1$ and $v_2$ and where $r^2 = r_1^2 + r_2^2$, then one has that

$$\langle Av, v \rangle = r_1^2 N \frac{p_1 + p_2}{2} + \mathcal{O}(\sqrt{N} + \frac{N}{\gamma}) f(r) + N \frac{p_1 - p_2}{2} r_2^2$$

for some function $f()$ such that $\|f\|_\infty \leq 1$. Thus it follows that $r_1 = 1 + \mathcal{O}(\frac{1}{\gamma} + N^{-\frac{1}{2}})$. This finishes the proof that the largest eigenvalue (and eigenvector) of $A$ and $P_0$ almost coincide. Similarly, since

$$\langle Av_2, v_2 \rangle = \lambda_2 + \mathcal{O}\left(\frac{N}{\gamma} + \sqrt{N}\right),$$

the same arguments imply that the second largest eigenvalue of $A$ and $P_0$ coincide provided

$$N(p_1 - p_2) \gg \sqrt{N} + \frac{N}{\gamma}.$$

414 And associated normalized eigenvectors coincide asymptotically, following the same basic perturba-
415 tion argument.

## C.2 Proof of Lemma 9

We first prove the first point. The objectif is to lower-bound $\langle v_1, w_1 \rangle$. Since $w_1$ has non negative coordinates and is normed to 1, $\sum_i w_1(i)|w_1|_\infty \geq 1 = |w_1|_2^2$. Thus we immediately get the first lower bound

$$\langle v_1, w_1 \rangle = \frac{1}{\sqrt{N}} \sum_{i=1}^{N} w_1(i) \geq \frac{1}{\sqrt{N}|w|_\infty}.$$

Let $i_o$ be a coordinate such that $w_1(i_0) = |w|_\infty$. Then one has that

$$\mu_1 w_{i_0} = \sum_{j=1}^{N} P_{i_0 j} w_j = \sum_{j=1}^{N} P_{i_0 j} w_{i_0} + \sum_{j=1}^{N} P_{i_0 j}(w_j - w_{i_0}).$$

417 Fix $\eta > 0, \epsilon > 0$ that we allow further to depend on $N$ and such that $\eta \gg \epsilon$. Using that $\mu_1 \geq$
418 $d_{\max}(1 - \epsilon)$ (see Proposition 2), we thus obtain that

$$\sum_{j=1}^{N} P_{i_0 j}(w_{i_0} - w_j) \leq \epsilon d_{\max} w_{i_0}, \tag{16}$$

where $d_{\max} = \max_i \sum_{j=1}^{N} P_{i,j} \simeq c_0 \frac{N}{\gamma}$. Define now

$$B := \{j, P_{i_0 j} > \eta \text{ and } w_j < \frac{w_{i_0}}{2}\}$$

and

$$\overline{B} := \{j, P_{i_0 j} > \eta \text{ and } w_\geq \frac{w_{i_0}}{2}\}$$

419 Using (16), one obtains that $\eta \sharp B w_{i_0}/2 \leq \epsilon w_{i_0} d_{\max}$. This means that

$$\sharp B \leq \frac{2\epsilon}{\eta} d_{\max}. \tag{17}$$

We can also deduce from the fact $\mu_1 \geq d_{\max}(1 - \epsilon)$ that

$$\sum_{j=1}^{N} P_{i_0 j} \geq d_{\max}(1 - \varepsilon).$$

Let us assume for the moment that

$$\sum_{j:P_{i_0 j} \geq \eta} P_{i_0 j} \geq c d_{\max}$$

for some constant $c$. Then by (17) this implies

$$\sharp\overline{B} \geq cd_{\max} - \sharp B \geq d_{\max}(c - \frac{2\varepsilon}{\eta}) \geq Cd_{\max}$$

for some constant $C > 0$. Using the fact that $\|w\| = 1$, this implies that $d_{\max}Cw_{i_0}^2/4 \leq 1$ which in turn yields that

$$|w|_\infty \leq \frac{C'}{\sqrt{d_{\max}}},$$

and then Lemma 9 will be proved.

Therefore, it remains to prove that

$$\sum_{j:P_{i_0j}\geq\eta} P_{i_0j} \geq cd_{\max}.$$

This is true if $i_0$ is such that $|X_{i_0}|^2 \leq \frac{\ln\gamma}{\gamma}$, by slightly adapting the proof of Lemma 3 and choosing $\eta$ of the order of $\min\{\sqrt{\varepsilon}, 1/\gamma\}$ – more precisely, the only change in the proof of Lemma 3, is the control of $S_1$.

One can easily extend this claim if $|X_{i_0}|^2 \leq \frac{K\ln\gamma}{\gamma}$ for some constant $K$ large enough. Now noting $\sum_j P_{ij}^*$ the subsum over those indices $j$ such that $P_{ij} \leq \eta$, one has that

$$\mathbb{P}\left(\exists i, |X_i|^2 \geq \frac{K\ln\gamma}{\gamma} \sum_j^* P_{ij} \geq d_{\max}(1 - 2\epsilon)\right)$$

$$\leq \mathbb{P}\left(\exists \frac{CN}{\gamma}\right) \text{ points } X_j \text{ in a ball } B(x,r), |x| \geq \frac{(K-1)\ln\gamma}{\gamma}, r \leq 2\frac{\ln\gamma}{\gamma}\right).$$

$$\leq C''\binom{N}{\frac{N}{\gamma}}e^{-C'N(K-2)\ln\gamma},$$

where $C, C', C''$ are constants and the last follows from Gaussian integration on squares of size $2\frac{\ln\gamma}{\gamma}$ covering $B(0, (K-3)\frac{\ln\gamma}{\gamma})^c$. Choosing $K$ large enough (actually $K = 4$ should be enough) yields the result and finishes the proof of the first part Lemma 9.

We now consider the second, more technical point. Let us consider a subset of indices $I \subset \{1, \dots, N\}$ to be fixed later and $w_I = \frac{1}{\sqrt{\sharp I}}(w_I(1), \dots, w_I(N))^t$, where $w_I(i) = \mathbb{1}_{i\in I}$.

Then one has

$$\langle w_I, v_1\rangle = \sqrt{\frac{\sharp I}{N}} \text{ and } \langle P_1 w_I, w_I\rangle = \frac{1}{I}\sum_{i,j\in I} P_{ij} =: D_I,$$

where $D_I$ denotes the average inner degree (restricted to edges between two vertices from $I$) and it also holds that $\langle P_1 v_1, v_1\rangle = \overline{d}$ where $\overline{d}$ is the average global degree. We now show that we can exhibit such a set $I$ such that $\sharp I \geq \gamma$ and $D_I = \mu_1(1 + o(1))$, since we assumed $\frac{N}{\gamma} \sim Np$. Fix $A > 0$. Set

$$I := \{1 \leq i \leq N, \|X_i\|^2 \leq A\frac{\gamma}{N}\}.$$

Since $\gamma\ln\gamma/N$ tends to 0, the arguments of the proof of Lemma 3 can be easily adapted to prove that $\sharp I \geq \gamma A$ with arbitrarily high probability as long as $A \ll \ln\gamma$. Moreover, adapting again the proof of Lemma 3 (controlling the sum $S_1$ defined there in a similar fashion since we can still approximate $\mathbf{n_k^{(i)}}$ by $\frac{N\varepsilon}{2\gamma}$ as $e^{-\frac{\|X_i\|^2}{2}}$ goes to 1), we obtain that $D_I = \mu_1(1 + o(1))$. We can do the same to define a vector supported on $\frac{N}{\gamma}$ coordinates instead of $\gamma$.

Consider now the largest entry of $w_1$: let $i$ be such that $w_i = |w_1|_\infty$. Let $\epsilon$ be fixed small so that $\mu_1 \geq \frac{N}{2\gamma}(1 - \epsilon)$. Let $J$ be the subset

$$J = \{j, w_1(j) \geq (1 - 3\epsilon)w_i\}.$$

Then, one has that $\sum_{j \in J} P_{ij} + (1 - 3\epsilon)(\sum_j P_{ij} - \sum_{j \in J} P_{ij}) \geq \frac{N}{2\gamma}(1 - \epsilon)$ from which one deduces that $\sum_{j \in J} P_{ij} \geq \frac{2}{3}\frac{N}{2\gamma}$. In particular this implies that $w_1$ cannot be localized on less than $\frac{N}{\gamma}$ coordinates (and is roughly equally spread on these coordinates). One can also show that the second block of largest entries of $w_1$ has size at least of order $\frac{N}{\gamma}$ and entries greater than $|w_1|_\infty(1 - 3\epsilon)^2$. Assume $w_1$ is localized on less than $\gamma$ coordinates so that $\langle w_1, w_I \rangle \to 0$.

In the same way we constructed $I$, one can construct at least $\gamma^2/N$ vectors $\hat{v}_i$ whose support are of size $A\frac{N}{\gamma}$ $A > 0$ chosen large enough, 2 by 2 disjoint such that

$$\langle \hat{v}_i, P\hat{v}_i \rangle \geq \frac{N}{2\gamma}(1 - \epsilon).$$

Let now $\tilde{w}_1$ be the vector whose coordinates are those of $w_1$ greater than $\eta|w_1|_\infty$, with $\eta > 0$ chosen small. Because $w_1$ is localized on less than $\gamma$ coordinates, the number of non zero coordinates of $\tilde{w}_1$ can be written $k\frac{N}{\gamma}$ for some $k \ll \frac{\gamma^2}{N}$. Let $\epsilon$ be such that $1 - 3\epsilon = \eta$, so that there must exist an index $\mathbf{i} \in J$ such that for some $\delta > 0$,

$$\sum_{j \notin J} P_{ij} \geq \delta \frac{N}{\gamma}.$$

This follows from the fact that $J$ corresponds to a subset of indices of the smallest of the $X_i$'s and the nearest neighbors cannot be all in $J$. Furthermore, for the same reason there exist at least $\delta'\frac{N}{\gamma}$ such indices $\mathbf{i}$. Indeed define for any vertex $j \in J$:

$$S_1(j) = \sum_{k \in J} P_{jk}; S_2(j) = \sum_{l \in J^c} P_{jl}.$$

One then has that

$$\frac{\mu_1}{S_1(j) + S_2(j)} \to 1, \forall j \in J.$$

In all cases one has that

$$\frac{\mu_1}{S_1(j) + S_2(j)} \geq 1 - \epsilon.$$

Fix $\delta > 0$ small. And set $E_\delta = \{j \in J, \frac{S_1(j)}{S_1(j)+S_2(j)} \in [\delta, 1 - \delta]\}$. We call $E_\delta$ the boundary of $J$. For any $i = 1, \ldots, kN\gamma^{-1}$ (corresponding to the non zero entries of $\tilde{w}_1$), consider the ball $B(X_i, \frac{1}{\gamma})$. It is colored green if $\frac{S_2(i)}{S_1(i)+S_2(i)} > 1 - \delta$. It is colored red $\frac{S_1(i)}{S_1(i)+S_2(i)} > 1 - \delta$. In all other cases, such a ball is colored blue[3] . One can note that the boundary corresponds to blue balls. We claim that there exists $\delta > 0$ small such that the edge $E_\delta$ is non empty and furthermore encircles an area in the order of $k\frac{N}{\gamma}$.

To prove this fact, one first remarks that there are green balls. This follows from the fact that we assume the size of the support of $w_1$ is negligible with respect to $\gamma$. There also exists at least one red ball. Indeed, consider the ball centered at $X_i$ where $w_i = |w_1|_\infty$. One then has that

$$\frac{\mu_1}{S_1(i) + S_2(i)} = \frac{S_1(i)}{S_1(i) + S_2(i)}a_1 + \frac{S_2(i)}{S_1(i) + S_2(i)}a_2,$$

where $a_1 S_1 = \sum_{k \in J} P_{ik}\frac{w_k}{w_i}$, $a_2 S_2 = \sum_{l \in J^c} P_{il}\frac{w_l}{w_i}$. One deduces that

$$\frac{S_1(i)}{S_1(i) + S_2(i)} \geq \frac{\frac{\mu_1}{S_1(i)+S_2(i)} - \eta}{a_1 - a_2},$$

where $\frac{\mu_1}{S_1(i)+S_2(i)} \leq a_1 \leq 1$. From this one deduces that

$$\frac{S_1(i)}{S_1(i) + S_2(i)} \geq 1 - \frac{\epsilon}{1 - \eta}.$$

Choosing $\eta > 0$ small enough ($\eta < 1/2$) yields that

$$\frac{S_1(i)}{S_1(i) + S_2(i)} \geq 1 - 2\epsilon \geq 1 - \delta$$

444 provided $\delta \geq 2\epsilon$. Consider two balls intersecting on more than one third of the total area of one ball.
445 This is the case if the center of the second ball is contained in the first one. They cannot be colored
446 green and red provided $2\delta < 1/3$. From this fact we deduce that there necessarily exists an interface
447 of blue balls surrounding the red balls. Now $J$ consists of indices corresponding to those in the area
448 encircled by the blue interface (up to an error in the proportion of $\delta$) and some more points which
449 are necessarily included in red balls centered at some point $X_j, j \in J$. Note that the proportion of
450 those points in $J$ and such red balls cannot exceed $\delta$. The minimal area $A$ to contain $kN\gamma^{-1}$ points
451 is in the order of $A \geq Ck\gamma^{-1}$ for some constant $C$. Now the total area covered by red balls with
452 some inside points in $J$ defines a domain $D$ whose area is at most in the order of $\frac{k}{\gamma}$. Among these a
453 proportion of at most $2\delta$ corresponds to points in $J$. From this we deduce that the area encircled by
454 blue balls is at least $cA$ for some constant $c < 1$. Thus one can find at least $K = (k\gamma)^{1/2}$ blue disks
455 whose support are pairwise disjoint and on the frontier of the domain.

456 As a consequence there exists at least one normalized vector $\hat{v}_i$ such that the supports of $\hat{v}_i$ and $\tilde{w}_1$
457 are disjoint. Calling $I_2$ the support of $\hat{v}_i$ one has that there exists a constant $c > 0$

$$R_{v_2} := \sum_{i \in J, \, j \in I_2} P_{ij} w_1(i) \frac{1}{\sqrt{\sharp I_2}} = \frac{\mu_1}{\sqrt{\sharp I_2}} \sum_{i \in I_2} w_1(i) \geq c \sqrt{\frac{N}{\gamma}} \eta |w_1|_\infty \mu_1. \tag{18}$$

Now we can construct at least $K$ such vectors whose support are pairwise disjoint by considering the
blue disks. We denote these vectors $\mathbf{v_1}, \dots, \mathbf{v_K}$. Let then set

$$v = \frac{\sum_{i=1}^{K} \mathbf{v_i}}{\sqrt{K}}.$$

Then because $\langle \mathbf{v_i}, P\mathbf{v_i} \rangle \geq \frac{N}{2\gamma}(1 - \epsilon)$, and (18) one can check that

$$\sup_r \langle rw_1 + \sqrt{1 - r^2}v, P\left(rw_1 + \sqrt{1 - r2}v\right) \rangle$$

is achieved for $r_0 < 1$ such that

$$\frac{r_0}{\sqrt{1 - r_0^2}} \geq \frac{\mu_1 - \frac{N(1-\epsilon)}{2\gamma}}{\sqrt{K}c\sqrt{\frac{N}{\gamma}}\eta|w_1|_\infty \mu_1}.$$

458 The denominator is much larger than $\mu_1$ as one can check that $\sqrt{K}\sqrt{\frac{N}{\gamma}}|w_1|_\infty$ does not tend to 0.
459 And furthermore this maximum can excede $\mu_1$: this is a contradiction.

460 ## C.3  Proof of Theorem 10

Let us denote by $\theta_1$ and $\theta_2$ the two eigenvalues that exit the support of the spectral measure of $P_1$.
Now assuming this holds true, an eigenvector associated to such an eigenvalue $\theta$ has necessarily the
form:
$$w = R_1(\theta)(\alpha_1 v_1 + \alpha_2 v_2),$$
where
$$\alpha_1 v_1 + \alpha_2 v_2 \in \text{Ker}(I + P_0 R_1).$$
461 Hereabove and in the sequel we denote $R_1$ for $R_1(\theta)$ for the sake of notations. Using this one deduces
462 that

$$\alpha_1 = -\frac{\lambda_1 \langle v_1, R_1 v_2 \rangle}{\lambda_1 \langle v_1, R_1 v_1 \rangle + 1}\alpha_2$$
$$\text{and } \lambda_1 \lambda_2 \langle v_1, R_1 v_2 \rangle^2 = (1 + \lambda_1 \langle v_1, R_1 v_1 \rangle)(1 + \lambda_2 \langle v_2, R_1 v_2 \rangle).$$

463 Then for such an eigenvector setting $a_i = \langle v_i, R_1 v_i \rangle$, for $i = 1, 2$ and $b = \langle v_1, R_1 v_2 \rangle$ we obtain that

$$\langle w, v_2 \rangle^2 = \frac{\alpha_2^2}{\lambda_2^2}; \langle w, v_1 \rangle = \frac{b\alpha_2}{1 + \lambda_1 a_1}. \tag{19}$$

So far we have not normalized the eigenvector $w$: this has to be considered in order to show that there is indeed some information on $v_2$ using the two normalized eigenvectors. Let us now recall the equation to compute the two eigenvalues $\theta_i$:

$$f_{\lambda_1,\lambda_2}(\theta) = (1 + \lambda_1 a_1(\theta))(1 + \lambda_2 a_2(\theta)) - \lambda_1 \lambda_2 b^2(\theta) = 0, \tag{20}$$

which we have solved as $\theta$ being a function of $\lambda_1$ and $\lambda_2$. The very definition of $w$ yields that

$$||w||^2 = \alpha_2^2 \left( \frac{\lambda_1^2 b^2}{(\lambda_1 a_1 + 1)^2} a_1'(\theta) + a_2'(\theta) - 2 \frac{\lambda_1 b}{\lambda_1 a_1 + 1} b'(\theta) \right).$$

Using (20) we obtain that

$$||w||^2 = \alpha_2^2 \frac{\frac{\partial f_{\lambda_1,\lambda_2}}{\partial \theta}}{\lambda_2(1 + \lambda_1 a_1)} = \alpha_2^2 \frac{\frac{\partial f_{\lambda_1,\lambda_2}}{\partial \theta}}{\frac{\lambda_1 \lambda_2 b^2}{a_2^2} - (1 + \lambda_1 a_1)}, \tag{21}$$

and combining (19) and (21) gives

$$\frac{\langle w, v_2 \rangle^2}{||w||^2} = \frac{1}{\frac{\partial f_{\lambda_1,\lambda_2}}{\partial \theta}} \frac{1 + \lambda_1 a_1}{\lambda_2}. \tag{22}$$

Notice that Equation (22) implies that there are at most two eigenvalues of $P_0 + P_1$ that separate from the spectrum of $P_1$; denote them by $\theta_1$ and $\theta_2$. We also recall that we have denoted by $\theta(\lambda_1)$ and $\theta(\lambda_2)$ the respective solutions of $1 + \lambda_1 a_1 = 0$ and $1 + \lambda_2 a_2 = 0$. We claim that those four specific values satisfy the following relations

$$\begin{aligned} \theta_2 &\le \min\{\theta(\lambda_2), \theta(\lambda_1)\} \\ \theta_1 &\ge \max\{\theta(\lambda_2), \theta(\lambda_1)\} \end{aligned}, \quad \theta(\lambda_2) \le \lambda_2 + \mu_1 \quad \text{and} \quad \lambda_1 \le \theta(\lambda_1) \le \lambda_1 + \mu_1.$$

The inequalities on the left are a consequence of the fact that $\theta_1$ and $\theta_2$ are solutions of $f_{\lambda_1,\lambda_2}(\theta) = 0$ thus $(1 + \lambda_1 a_1(\theta_i))$ and $(1 + \lambda_2 a_2(\theta_i))$ must have the same sign, the one of $\frac{\partial f_{\lambda_1,\lambda_2}}{\partial \theta}(\theta_i)$. The second inequality is a consequence of the fact that $|\mu_j| \le \mu_1$ and then plugging this value in $a_2$. The inequalities on the right are a consequence of the very last argument and of the fact that $\theta(\lambda_1) \ge \lambda_1$ since $\theta(\lambda_1)$ is an eigenvalue of $P_0 + \lambda_1 v_1 v_1^\top$.

This immediately gives the first bound

$$-(1 + \lambda_1 a_1(\theta_2)) = \lambda_1 \sum_j \frac{r_j^2}{\theta_2 - \mu_j} - 1 \ge \frac{\lambda_1}{\lambda_2 + 2\mu_1} - 1 \tag{23}$$

As a consequence, it remains to control $\frac{\partial f_{\lambda_1,\lambda_2}}{\partial \theta}(\theta_2)$. Notice that, by definition of $f_{\lambda_1,\lambda_2}$ and the fact that $f_{\lambda_1,\lambda_2}(\theta_2) = 0$, we get

$$\left| \frac{\partial f_{\lambda_1,\lambda_2}}{\partial \theta}(\theta_2) \right| \le \lambda_1 \frac{\partial a_1}{\partial \theta}(\theta_2)(\lambda_2|a_2| - 1) + \lambda_2 \frac{\partial a_2}{\partial \theta}(\theta_2)(\lambda_1|a_1| - 1)$$

$$+ 2 \frac{\partial b}{\partial \theta}(\theta_2) \sqrt{\lambda_1 \lambda_2} \sqrt{(1 + \lambda_1 a_1)(1 + \lambda_2 a_2)}$$

Moreover, we immediately get the following upper-bounds

$$|a_i(\theta)| = \sum_j \frac{r_j^2}{\theta - \mu_j} \le \frac{1}{\theta - \mu_1}, \quad |a_2(\theta)| \le \frac{1}{\theta - \mu_1}, \quad a_1', a_2', b' \le \frac{1}{(\theta - \mu_1)^2}.$$

Plugging those estimates in $\frac{\partial f_{\lambda_1,\lambda_2}}{\partial \theta}(\theta_2)$ gives that

$$\lambda_2 \left| \frac{\partial f_{\lambda_1,\lambda_2}}{\partial \theta} \right| \le \frac{\lambda_1 \lambda_2}{(\theta_2 - \mu_1)^2} \left( \frac{\lambda_2}{\theta_2 - \mu_1} - 1 \right) + \frac{\lambda_2^2}{(\theta_2 - \mu_1)^2} \left( \frac{\lambda_1}{\theta_2 - \mu_1} - 1 \right)$$

$$+ 2 \frac{\sqrt{\lambda_1 \lambda_2} \lambda_2}{(\theta_2 - \mu_1)^2} \sqrt{\left( \frac{\lambda_2}{\theta_2 - \mu_1} - 1 \right) \left( \frac{\lambda_1}{\theta_2 - \mu_1} - 1 \right)} \tag{24}$$

From Equation (5), we get that $\theta_2 \geq \frac{\lambda_2}{4} \geq \mu_1(1+\varepsilon)$ so that we get non-zero correlation between $w_2$ and $v_2$ from Equations (23) and (24).

We can actually be more precise. It is indeed quite easy to prove using (**??**) that

$$f_{\lambda_1,\lambda_2}(\theta) \geq 1 + \frac{\lambda_1}{\mu_1 - \theta} + \frac{\lambda_2}{\mu_1 - \theta} + \frac{\lambda_1\lambda_2}{(\mu_1 + \theta)^2}.$$

Let us assume that the ratios $\frac{\lambda_1}{\lambda_2} = q > 1$ and $0 \leq \frac{\mu_1}{\lambda_2} = x \leq 1$ are fixed, and make the change of variables $\theta = \lambda_2 - \gamma\mu_1 = (1 - \gamma x)\lambda_2$, so that

$$f_{\lambda_1,\lambda_2}(\theta) \geq 1 - \frac{1+q}{1-(\gamma+1)x} + \frac{q}{(1-(\gamma-1)x)^2}.$$

In order to control the solution of $f_{\lambda_1,\lambda_2} = 0$ w.r.t. $\gamma$, we are going to assume for the moment that $(\gamma+1)x \leq \frac{1}{2}$ so that the r.h.s. can be easily lower-bounded into

$$
\begin{aligned}
f_{\lambda_1,\lambda_2}(\theta) &\geq 1 - (1+q)\big(1 + (\gamma+1)x + 2((\gamma+1)^2 x^2)\big) \\
&\quad + q\big(1 + 2(\gamma-1)x - (\gamma-1)^2 x^2\big) \\
&= x\Big(\big[\gamma(q-1) - (3q+1)\big] - 2x\big[(3q+1)\gamma^2 - 2(q-1)\gamma + (3q+1)\big]\Big),
\end{aligned}
$$

which gives an explicit (and uniformly bounded) upper-bound $\overline{\gamma}$ for $\gamma$, i.e., the solution of the above degree 2 polynomial. Notice that when $x$ goes to zero, the expression boils down to

$$\overline{\gamma} = 3 + \frac{4}{q-1} + \mathcal{O}(x).$$

Plugging $\overline{\gamma}$ into Equations (23) and (24) gives that

$$\frac{|\langle w, v_2\rangle|^2}{\|w\|^2} \geq \Big(1 - \frac{2x}{q-1}\Big)\frac{(1 - (\overline{\gamma}+1)x)^3}{\Big(1 + \frac{\overline{\gamma}+1}{2(q-1)}x + \sqrt{q(\overline{\gamma}+1)x}\Big)^2}$$

which is uniformly bounded away from 0.

Moreover, when $x$ goes to 0, it holds that

$$
\begin{aligned}
\frac{|\langle w, v_2\rangle|}{\|w\|} &\geq 1 - 2\frac{q}{\sqrt{q-1}}\sqrt{x} - \mathcal{O}(x) \\
&= 1 - 2\frac{\frac{\lambda_1}{\lambda_2}}{\sqrt{\frac{\lambda_1}{\lambda_2} - 1}}\sqrt{\frac{\mu_1}{\lambda_2}} - \mathcal{O}\big(\frac{\mu_1}{\lambda_2}\big)
\end{aligned}
$$

and when $x$ is small enough[4], then we also have that $(\gamma+1)x \leq \frac{1}{2}$ as required. This proves the theorem (since ratios are assumed to be uniformly lower and upper-bounded).

## Footnotes

[3]Of course, this choice of colours is completely arbitrary and only for illustration purpose

[4]Numerical implementation suggests that those computations hold for $x \leq \frac{q-1}{8q}$, i.e., when the value on $\overline{\gamma}$ is set to $3 + \frac{4}{q-1}$ without the $\mathcal{O}(x)$ term.