[Reviews · NeurIPS 2020]

Review 1

Summary and Contributions: The paper discusses graph partitioning of the 2-block stochastic block model (SBM) when the realized graph includes edges that are drawn not only from the SBM model but also from a random geometric model. The combination of the two models makes sense in practice since a lot of large-scale social networks include edges that are explained by endogenous features (geometric model) as well as exogenous features (SBM) of the nodes. The authors are interested in recovering the clusters that are created due to exogenous features using the second eigenvector of the adjacency matrix. The authors show exact and weak recovery guarantees of the clusters that are defined by the 2-block SBM. To the best of my knowledge, the results are novel.

Strengths: * The paper is theoretical sound. The exact and weak recovery results that are presented are novel and are closer to what we observe in practice.

Weaknesses: * Although the paper is solid theoretical and interesting work. I am not certain if this paper is relevant to the NeurIPS community. I think this paper is best suited for theoretical statistical venues. * Only the 2-block case is considered, which makes the results less interesting.

Correctness: yes

Clarity: The paper is well written.

Relation to Prior Work: Yes

Reproducibility: Yes

Additional Feedback: Update after rebuttal: Thank you for your reply. I decided to keep my overall score unchanged. ================== I do not have any major comments. The paper is clearly written and solid. * One minor comment is that the writing style is very dry and this makes the paper difficult to read. * I think it would be good to discuss assumptions that are made about the relation of p_1, p_2, N, gamma, kappa. * Also, please provide certain settings of the parameters that you think are reasonable before you provide the result. This will provide some context to the reader and will ease understanding of the paper.


Review 2

Summary and Contributions: The paper deals with the community detection for networks generated from the stochastic block model (SBM) perturbed with errors from random geometric graphs (RGG). The paper considers that the probability of edge formation in the SBM is perturbed by an error, which is a radial function of the distance between the latent embeddings of the nodes in a two-dimensional Euclidean space. The new model has the variance parameter of the radial kernel as an extra parameter along with the parameters of the SBM. The main contributions of the paper are - (1) Deriving a theoretical result for spectral properties of RGG with radial kernel. (2) Deriving theoretical results for SBM perturbed with noise as RGG.

Strengths: Soundness of claim: The paper provides rigorous theoretical justification for the claims made in the paper. The proofs of the two main results, one on random geometric graphs and another on SBM perturbed by RGG are given in the Appendix. However, some of the proof structure and intermediate results are represented in the paper in the form of Proposition and Lemma. Significance and novelty: The paper is significant and novel in terms of giving a theoretical results on spectral structure of RGG and spectral structure of SBM with errors in terms of RGG noise. The paper extends the current knowledge of community detection for SBM and RGG. Relevance: The results presented in the paper are relevant to the networks community, as it gives a basic framework based on which further theoretical and practical studies will be possible in presence of covariates and latent variables within RGG.

Weaknesses: Soundness of claim: The paper provides proof structure of the theoretical results in the main paper, but the simulation study is relegated to the Supplement. Significance and novelty: The paper has significant results, but the main mathetical toolbox for the proofs already exist and are in frequent use in the current literature. It would be better if the paper had gone for regimes sparser than degree of log(n), as the regime of 1/gamma > log(n)/n is interesting but 1/gamma < log(n)/n regime can become more significant and might have to involve novel technique. Relevance: The paper is relevant but some follow-up discussions, such as, use of covariates in place of latent variable based noise for analysis, might help in follow-up works.

Correctness: The paper mostly relies on rigorous proofs of the theoretical results. I only went through the proofs in Supplement cursorily but the proof steps as well as the results seemed intuitive and correct.

Clarity: The paper is mostly well-written but there are some typos and grammatical errors like in lines 80, 132, 185.

Relation to Prior Work: The paper builds up on the current literature on spectral properties of SBM and RGG. The paper draws from the existing proof techniques on eigenvalue bounds for random matrices but has sufficiently referred the relevant works to my knowledge.

Reproducibility: Yes

Additional Feedback: There are quite a few typos and writing issues in the paper. They need to be addressed in the paper, if accepted.


Review 3

Summary and Contributions: This work provides theoretical analysis of spectral methods to do community detection over the graphs that hold topology generated by stochastic block model while perturbed by random geometric variables. This work discussed the impact of different regimes of the model parameters on the algorithm.

Strengths: (1) It is novel and interesting to consider the community detection problem with this new model SBM + perturbation from random geometric latent variables. (2) The analysis reads rigorous and the logic is clear. I enjoy reading it.

Weaknesses: (1) The template is weird. There is no section number before introduction. (2) Although this is a theoretical work, it is not good to miss the total section for empirical evaluation. At least, the dependence on the parameters of the obtained regime should be demonstrated via simulation. --- Thank the authors for preparing the response. I think this work has solid theory and I lean to accept this work. However, just as previously argued, more intuition should be provided and detailed proof can be postponed to the supplement. Moreover, more numerical evaluation is needed before I can increase my overall evaluation.

Correctness: I do not check the proof in details. However, I think the arguments are sound

Clarity: Although I have published 1-2 papers in this direction, this paper is here and there hard to follow. Readers need to know much background along the research of this work. It is good to give more exposure of relevant literature.

Relation to Prior Work: The position of this work is clearly discussed.

Reproducibility: Yes

Additional Feedback:


Review 4

Summary and Contributions: The authors consider random graphs with nodes partitioned into two equal-sized communities, and where edges are put randomly with a probability that is a sum of a community-dependent term and a geometry-dependent term. The geometry part is defined through a gaussian kernel, and 2-diemensional node features themselves iid gaussian. They ask whether spectral methods based on the adjacency matrix of such graphs could enable non-trivial reconstruction of the underlying partition into two communities. They provide positive answers in specific parameter ranges.

Strengths: Little work has addressed robustness of spectral clustering methods to the types of pertubations considered in this paper.

Weaknesses: The model is very specific, and so the results are of limited applicability.

Correctness: The results appear correct.

Clarity: There are many typos. Some statements are unclear.

Relation to Prior Work: Yes. post-rebuttal fruther input: http://proceedings.mlr.press/v99/stephan19a.html which addresses robustness of spectral methods to adversarial perturbations, covering a sparse degree setup with an arbitrary number of communities in the SBM, appears to be highly relevant prior work.

Reproducibility: Yes

Additional Feedback: -on page 4, the term "degree" is used improperly, with reference to matrix P. A clearer terminology would be needed. -page 6: det(I+P_0R_1) is not a polynomial. Function theta(.) mentioned on line 172 has not been introduced previously in the text. post-rebuttal addition: I still feel that the paper is a little bit below the threshold: again it analyses a very specific model. And it would be much more interesting if it highlighted the key properties verified in this specific model that allows one to obtain positive results, thus making it more obvious to the reader how to extend the arguments to other models. Also, the writing is not fully satisfactory, with many typos (lines 25, 59, 63, 80, 132 --on line 138 it should mention that independence is only up to symmetry--, line 169 requires some rewording, in line 172 it doesn’t say where this theta function for the rank one case has been introduced).

[Author Response · NeurIPS 2020]

First of all, we would like to thank the reviewers for their positive feedbacks; you all mentioned the novelty, the relevancy and the soundness of our results.

**More discussions and dry writing** It seems that the major common complaint is the lack of discussions and the "dry" writing. We apologize for that, but this is mostly due to the NeurIPS page limit (otherwise, we would have had to postpone almost all the proofs and insights to the Appendix. Luckily, we will have one extra page of discussions in the revised version – if the paper is accepted). Your suggestions to gain space are also more than welcome: maybe we could remove the proof of Lemma 7 (it is not always clear what brings intuitions to the reader) ?

This extra page will give us room to discuss the relation between the different parameters and why this or that regime is interesting.

**A sparser regimes would be interesting.** We 100% fully agree with you that studying sparser regimes is of utmost interest. This is definitely future work; notice that this paper is already 26 pages long (and many discussions should be added to give a better understanding!), this is why we believed the actual version of this paper is already quite interesting (as almost all of you said).

We think this is actually a good sign. It shows that this line of work is just open and it will generate follow-up papers.

**Only the 2-block model is consider.** As above, going beyond 2-block is also left for future work. We had to start by the simple and standard SBM model before generalizing our robustness results. First computations seem to indicate that it is indeed possible to extend our approach to more than 2 blocks, under some relatively strong assumptions (like strong balancedness of all communities and other technical details); but totally satisfactory results requires more work.

**More simulations would be appreciated.** It's true that we only provide, in the Appendix, "one" simulation (i.e., for one set of parameters) that illustrate the dependency in $\gamma$. We will run and add several other simulations to show how the different parameters interplay in practice (we also did not include simulations illustrating the spectrum shape of the random graph - we will add them to the Appendix). That's a great suggestion, thanks.

**The model is specific and the results of limited applicability** We respectfully disagree. SBM are quite standard and very well studied by different communities. The perturbations we consider are generated via some geometric graph which also sounds quite general to us. Maybe you disagree with our specific choice of the Gaussian kernel, but, as we mention in the text, many other standard kernels could have been used (one just need to redo to proofs and computations to find different constants - but this is merely an exercice). Similarly, the fact that the geometric graph is in 2D is also irrelevant. It can be generalized (again, at the cost of intensive and, we think, without real additional interest computations).

This said, we agree that considering other types of robustness like the "worst-case" one (the detection algorithm should work with any distribution in a $\varepsilon$-ball around the standard SBM one) is also interesting. As before, this is left for future work (and, hopefully, to many follow-up papers).

**The term degree is used improperly** Thanks, we may use "connectivity" instead.

Thanks again for your feedbacks and constructive reviews. We think we answered your major concerns in this rebuttal and hope that, along with the other positive reviews, it will help you reassess positively your scores.

[Meta-Review · NeurIPS 2020]

The authors consider the problem of community detection in a two class SBM whose edges are perturbed by a latent random geometric graph. The theoretical results show that the second eigenvector of adjacency matrix is highly correlated to the latent community membership vector under a broad parameter regime. The reviewers mostly feel that this is interesting work and I will recommend accepting this paper. However the authors should find a way to move part of the simulation study to the main paper, and address other typographic issues pointed out by the reviewers. While reviewer 4 had the most negative view of this paper, he/she does not object to acceptance. In line with his/her remark, I will strongly urge the authors to cite “Robustness of Spectral Methods for Community Detection” since it is extremely relevant to this work.